# Poly-β-D-(1,6)-N-acetyl-glucosamine (PNAG) glycan vaccines with broad spectrum neutralizing activities

Kuo-Shiang Liao[1,2], Mu-Rong Kao[1,2,3], Tzu-Hsuan Ma[2,3], Mei-Hua Hsu[4], Tzu-Yin Chen [2], Balázs Imre[1,2,3], Philip J. Harris [5], Jiun-Jie Shie [6], Cheng-Hsun Chiu[4], Chung-Yi Wu[1] & Yves S. Y. Hsieh [1,2,3] ✉

The development of bacterial vaccines is a complex challenge due to the substantial serological diversity of protective antigens. One promising antigenic target is the conserved surface polysaccharide poly-β-(1,6)-N-acetyl-D-glucosamine (PNAG). Despite its widespread distribution, antibodies raised against PNAG have shown restricted efficacy in promoting microbial elimination in vitro and safeguarding against infections in vivo. Systematic studies and vaccine development have been hindered by limited knowledge of optimal antigenic features, such as chain length and degree of N-acetylation. Here, we describe an effective $n + 2$ glycosylation strategy enabling controlled synthesis of partially (dPNAG) and fully deacetylated PNAG glycans. Glycan microarray analysis shows that dPNAG glycans with DP8 and DP12 are optimal, with corresponding protein conjugates eliciting the highest IgG titers. Sera containing antibodies against the dPNAG DP8 conjugate with 40% acetylation exhibit the best opsonic activity against three prevalent nosocomial pathogens and confer the highest protection in female BALB/c mice against *Staphylococcus aureus*, supporting its potential as a vaccine candidate.

Microbial pathogens have emerged as the primary cause of a significant number of life-threatening, hospital-acquired (nosocomial) infections[1,2]. To combat these pathogens, two common approaches are the use of antibiotics and preventive vaccination. Indiscriminate treatment with antibiotics, however, results in enhanced antibiotic resistance[3]. Vaccines are being developed to combat certain resistant bacteria, including methicillin-resistant *Staphylococcus aureus* (MRSA). However, the design of vaccines with a broader and more effective protection against multiple resistant bacterial species is urgently needed[4–8].

Poly-β-(1,6)-N-acetyl-D-glucosamines (PNAGs) are linear polysaccharides (exopolysaccharides) that occur in the surface capsules of many pathogenic bacterial species as well as in associated biofilms[9].

PNAGs prepared from *Staphylococcus aureus* cell walls have been exploited as potential immune antigens in vaccine development[10–13] Both PNAG conjugated to proteins as vaccine candidates[11,14–22] and monoclonal antibodies raised against PNAGs[23–27] have shown protection against *S. aureus* in both in vivo and in vitro models[28,29]. Recent advances in the development of synthetic PNAG-conjugated with tetanus toxoid vaccine AV0328 by the US-based biotech company Alopexx, with the completion of Phase I human clinical trials, highlight PNAG's potential as a therapeutic target for combating multidrug-resistant infections and biofilm-associated diseases.

Indeed, PNAGs are conserved antigenic targets in a variety of pathogenic bacteria[30–32]. Because PNAGs are insoluble polymers, they

[1]Genomics Research Center, Academia Sinica, Taipei, Taiwan. [2]School of Pharmacy, College of Pharmacy, Taipei Medical University, Taipei, Taiwan. [3]Division of Glycoscience, Department of Chemistry, School of Engineering Sciences in Chemistry, Biotechnology and Health, KTH Royal Institute of Technology, AlbaNova University Centre, Stockholm, Sweden. [4]Molecular Infectious Disease Research Center, Chang Gung Memorial Hospital, Chang Gung University College of Medicine, Taoyuan, Taiwan. [5]School of Biological Sciences, The University of Auckland, Auckland Mail Centre, Auckland, New Zealand. [6]Institute of Chemistry, Academia Sinica, Taipei, Taiwan. ✉e-mail: yvhsieh@kth.se

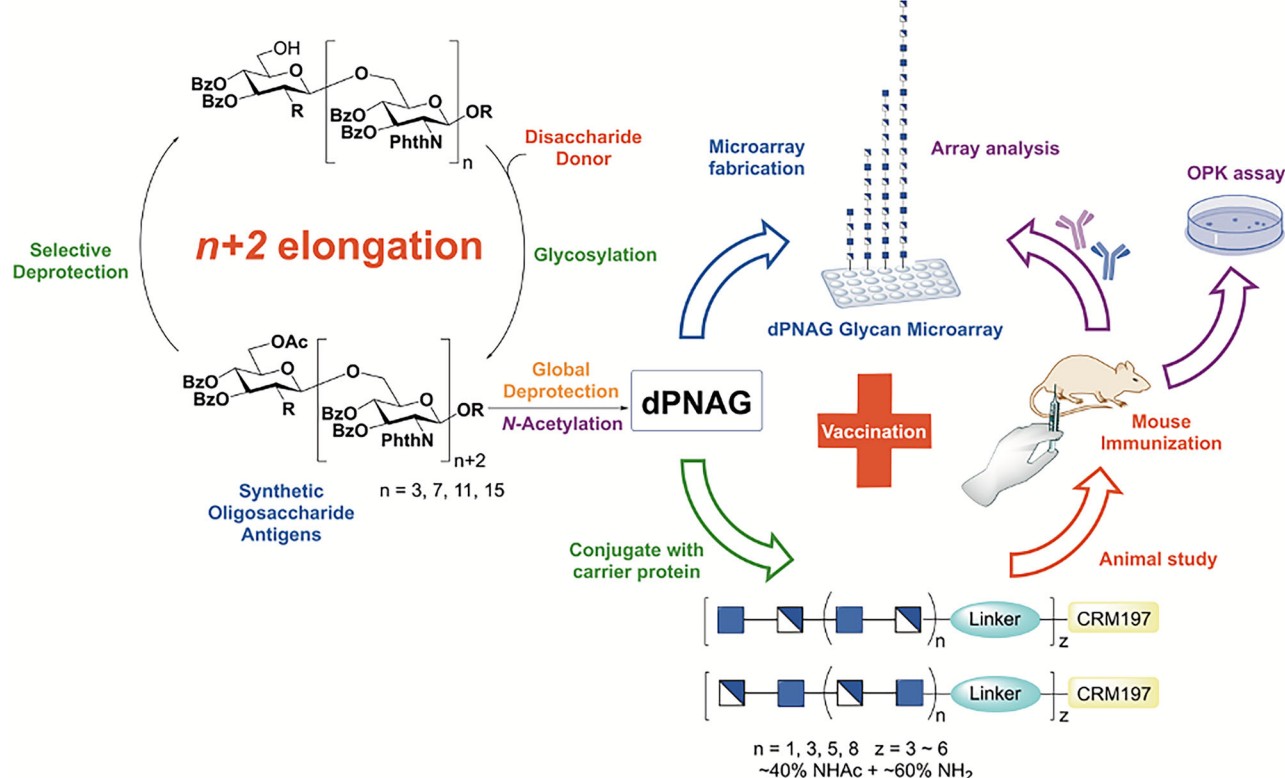

**Fig. 1 | Overview of the synthesis and immunological evaluation of dPNAG-CRM197 glycoconjugates.** Oligo-β-(1 → 6)-D-glucosamine derivatives (DP4–18) were synthesized via an efficient *n* + 2 glycosylation strategy using a *N*-phthaloyl glucosamine disaccharide thiocresol-based donor. Controlled acetylation yielded dPNAG glycans with defined degrees of polymerization and 40–60% *N*-acetylation. These glycans were used to construct glycan microarrays, which detected anti-dPNAG antibodies in sera from patients infected with *A. baumannii* and *S. pneumoniae*. In a separate study, selected glycans were conjugated to CRM197 and used to immunize mice, eliciting robust IgG responses. Immunogenicity and functional activity of the antisera were assessed by glycan arrays and opsonophagocytic killing (OPK) assays.

are difficult to study in vitro. However, soluble variants of PNAGs, for example, partially deacetylated PNAG (dPNAG) glycans, have shown potential as protective agents in animal studies, showing a notable 15-50% enhancement compared with the PNAG polymer. Nevertheless, the currently used preparation of dPNAG glycans is still far from ideal, particularly in its limited ability to induce potent opsonic killing and immune protection against PNAG-producing bacteria[33,34]. Our understanding of how factors such as the degree of polymerization (DP), the degree of *N*-acetylation, and the pattern of *N*-acetylation influence their effectiveness remains limited. Thus, the synthesis and optimization of the structures of dPNAG glycans have the potential to lead to broader protection against resistant microbial pathogens.

In the present study, we devised an efficient *n* + 2 elongation strategy using an *N*-phthaloyl glucosamine disaccharide thiocresol-based donor (**10**) to synthesize only the β forms of oligo-β-(1,6)-*N*-phthaloyl-D-glucosamine derivatives, ranging from tetramers (4mer) to octadecamers (18mer) (Fig. 1). The facile protection and deprotection of the acceptor hydroxyl group at the C-6 position of glucosamine was achieved by simple acetylation and deacetylation; moreover, the differences in polarities of -OH and -OAc offers a simple, straightforward way of separation on a flash column and by thin layer chromatography (TLC). Following global deprotection, by carefully regulating the proportion of the acetylating agent, we successfully generated partially *N*-acetylated dPNAG glycans (**41–49**) with defined DPs and degrees of acetylation. The 40–60% deacetylation range was selected because previous studies predominantly focused on either fully acetylated or non-acetylated oligomers. These glycans were used to construct glycan arrays, which were initially used to demonstrate that patients infected with the pathogenic bacteria *Actinetobacter*

*baumannii* (serotype 17978) and *Streptococcus pneumoniae* (serotypes 19 A) produced antibodies that recognized dPNAG glycans. Subsequently, the partially *N*-acetylated dPNAG glycans were conjugated to the diphtheria toxin mutant CRM197, a derivative of DT, to elicit strong T-cell-dependent immune responses in a mouse model (Fig. 1). The glycan arrays were further used to evaluate the immunogenicity of dPNAG-CRM197 conjugates and to identify the structures that gave the highest IgG titers. The mouse antisera were also used to suppress the growth of pathogenic bacteria, with their effectiveness being measured by an opsonophagocytic killing (OPK) assay (Fig. 1). Our study demonstrated the systematic immunological screening of panels of dPNAG-CRM197 conjugates as vaccine candidates with longer chain lengths and optimal levels of *N*-acetylation. These candidates protect against a broad spectrum of bacterial pathogens.

## Results

### Assembly of dPNAG glycans and their derivatives via a *n* + 2 glycosylation strategy

A series of dPNAG glycans with various defined lengths and degrees of *N*-acetylation were synthesized. First, we devised a strategy based on D-glucosamine hydrochloride as a starting material to synthesize the glucosamine monosaccharide donor and acceptor building blocks **6, 7, 8,** and **9** (Fig. 2). Next, glycosylation of the thioglycoside acceptor **6** and the glycosyl phosphate donor **8** provided disaccharide **10** in 99% yield as a pure β-stereoisomer. We also obtained the pure β-isomer product **11** in 91% yield using **7** as a donor and **9** as an acceptor; then, the selective removal of the only *O*-acetyl group on **11** by treatment with AcCl in MeOH afforded the terminal acceptor disaccharide **12** (Fig. 3).

**Fig. 2 | Synthesis of glucosamine monosaccharide building blocks 6, 7, 8, and 9.** Reagents and conditions: (a) NaOMe, phthalic anhydride, MeOH, 65 °C, 2 h; (b) Ac₂O, pyridine, RT, o/n, 73% (two steps); (c) TolSH, BF₃O·Et₂O, CH₂Cl₂, RT, 24 h, 65%; (d) NaOMe, MeOH, RT, 2 h; (e) TrCl, cat. DMAP, pyridine, 50 °C, o/n, 95%, (two steps); (f) BzCl, cat. DMAP, pyridine, RT, o/n, 88%; (g) *p*TSA, MeOH/CH₂Cl₂, RT, 8 h,

91%; (h) Ac₂O, pyridine, RT, 2 h, 99%; i HOPO(OBu)₂, NIS/TfOH, CH₂Cl₂, MS 4 Å, 0 °C, o/n, 95%; (j) HO(CH₂)₅N₃, TMSOTf, CH₂Cl₂, MS 4 Å, −40 to −25 °C, 2 h, 90%; and (k) AcCl, CH₂Cl₂/MeOH, RT, o/n, 97%. Phth Phthalimide, Tr Trityl, DMAP 4-(Dimethylamino)pyridine, *p*TSA *p*-Toluenesulfonic acid monohydrate, NIS *N*-Iodosuccinimide.

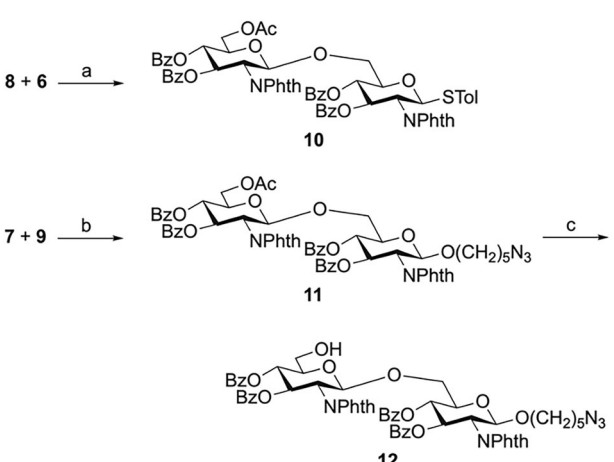

**Fig. 3 | Synthesis of disaccharide building blocks 10-12.** Reagents and conditions: (a) TMSOTf, CH₂Cl₂, MS 4 Å, −30 to 0 °C, 2 h, 99%; (b) NIS/TfOH, CH₂Cl₂, MS 4 Å, −20 °C, 95%; and (c) AcCl, CH₂Cl₂/MeOH, RT, o/n, 96%.

Fourteen dPNAG glycans of various lengths were synthesized via an iterative *n + 2* glycosylation and deprotection strategy using disaccharide **10** as the common donor. Glycosylation of the disaccharide donor **10** and acceptor **12** in the presence of NIS/TfOH in CH₂Cl₂ at −40 °C for 1 h gave the tetrasaccharide **13** as a pure β-isomer in 84% yield. The reducing-end building block **14** was obtained in 82% yield by removing the only *O*-acetyl group of the tetrasaccharide by treatment with AcCl in MeOH. Using the above procedures, the glycosylation of donor **10** to the alcohol acceptors **14, 16, 18, 20, 22, 24**, and **26** were repeated, activated by TMSOTf in CH₂Cl₂ at −30 to −20 °C for 2 h, allowing the assembly of the fully protected oligosaccharides **15, 17, 19, 21, 23, 25**, and **27** in 57%, 59%, 56%, 45%, 43%%, 48%, and 59% yields, respectively. After chromatographic separation, the glycosylated *O*-acetyl ester compounds were recovered, and deacetylation was repeated to yield the alcohol acceptors **16, 18, 20, 22, 24**, and **26** in 88%, 82%, 87%, 88%, 88% and 88% yields, respectively (Fig. 4).

Next, the azido-oligosaccharides were reduced with hydrogen using as a catalyst Pd (OH)₂ on activated charcoal to yield the amine compound which was then treated with thio-linker **a** in Et₃N to obtain the thiol-functionalized oligosaccharides **28**–**31**. After the global deprotection of the protected glycans (resulting in compounds **32**–**35**), the amino groups were partially acetylated to obtain thiol-functionalized linker oligosaccharides **36**–**40**. The degree of acetylation (-40 to -60%, determined by ¹H NMR) was controlled by adjusting the relative amount of the Ac₂O acetylating agent under weak basic conditions. Finally, the trityl group was removed by TFA and Et₃SiH to obtain the dPNAGs **41**–**49** with thiol-functionalized antigens for subsequent conjugation (Fig. 5).

**Evaluation of human antisera using dPNAG glycan microarrays**
The synthetic dPNAG glycans **41**–**49** with thiol-functionalized linkers were printed on maleimide-coated glass slides. The arrays were then used to profile the anti-dPNAG antibodies in sera from six patients infected with *A. baumannii* or with *S. pneumoniae* (serotypes 19 A, 19 F, or 23 F). Before doing the profiling, serum samples were selected based on the results of their enzyme-linked immunosorbent assay (ELISA) antibody titers. Our study showed significant elevations in anti-dPNAG IgG titers in two serum samples from *A. baumannii*-infected patients. This observation strongly indicated that all of the synthetic dPNAG glycans we produced were readily recognized by the host immune system (Fig. 6A, B). The binding profiles showed higher fluorescence intensities for the longer dPNAG glycans with partial *N*-acetylation, indicating the latter are better recognized. The induced IgG bound particularly strongly to the 18mer with 40% NHAc and the 12mer with 45% NHAc. Importantly, a clear opposite trend was observed in the binding profiles of the non-acetylated PNAG glycans, with shorter PNAG glycans being preferred over longer ones (4_NH₂ > 8_NH₂ > 12_NH₂ > 18_NH₂) (Fig. 6A, B). The glycan array study clearly confirmed that the degree of *N*-acetylation is a critical factor in determining PNAG immunogenicity. The antisera showed weaker binding affinities to the 12mer with 60% NHAc than to the one with 45% NHAc, while a strong response was also observed for the 18mer with 45% *N*-acetylation, suggesting that glycans with lower NHAc

**Fig. 4 | Stepwise synthesis of the glycans.** Reagents and conditions: a NIS/TfOH, $CH_2Cl_2$, −30 to −20 °C, 2 h **13**: ($n = 1$) 84%, **15**: ($n = 3$) 57%, **17**: ($n = 5$) 49%, **19**: ($n = 7$) 56%, **21**: ($n = 9$) 45%, **23**: ($n = 11$) 43%, **25**: ($n = 13$) 48%, **27**: ($n = 15$) 59%; and b AcCl, $CH_2Cl_2$/MeOH, RT, 1–2 d **14**: ($n = 3$) 82%, **16**: ($n = 5$) 88%, **18**: ($n = 7$) 82%, **20**: ($n = 9$) 87%, **22**: ($n = 11$) 88%, **24**: ($n = 13$) 88%, **26**: ($n = 15$) 88%.

ratios might be preferable antigens eliciting stronger humoral responses.

In addition, sera from four patients infected with *S. pneumoniae* were analyzed by glycan microarray (Fig. 6C–F). Two patients were infected with *S. pneumoniae* (serotype 19 A), one with *S. pneumoniae* (serotype 19 F), and one with *S. pneumoniae* (serotype 23 F). Consistent with the results obtained for *A. baumannii*, the binding strength gradually decreased as a function of the DP of the non-acetylated glycans, whereas the opposite was observed in the case of partially acetylated glycans. In a similar way, the 12mer with a lower degree of acetylation (45%) consistently induced stronger binding than the one with 60% NHAc. Critically, the glycan array study highlighted the importance of the degree of *N*-acetylation, providing valuable insights that may serve as a crucial reference in PNAG vaccine design.

## Synthesis and characterization of dPNAG-CRM197 conjugates

The conjugation of dPNAG glycans to a carrier protein enhances the immunogenicity of the glycans and assists T-cells in inducing somatic mutations and class switching to dPNAG-specific antibodies. Therefore, the synthetic dPNAG glycans **41–49** were conjugated to the nontoxic diphtheria toxin mutant CRM197 (DT) (Fig. 7)[11,35,36]. CRM197 is a FDA approved carrier protein that has been used in many licensed vaccines. The commercially available compound succinimidyl 3-(bromoacetamido)propionate (SBAP) was used as a coupling agent to generate a thiol-reactive 2-bromoacetyl group on CRM197 before the dPNAG glycans **41–49** were conjugated to CRM197 via a reaction between the 2-bromoacetyl and sulfhydryl groups to form stable, covalent thioether bonds. Finally, cysteine was used to quench the unreacted 2-bromoacetyl groups, resulting in the formation of glycan-CRM197 vaccine candidates **50–58** (Fig. 7). The relative proportions of glycan and carrier protein in the conjugates were determined by MALDI-TOF MS (Table S1).

## Immunogenicity assessment of dPNAG-CRM197 conjugates in mice using glycan microarrays

The immunogenicity of dPNAG-CRM197 conjugates was evaluated by immunizing 6–8-week-old, female BALB/c mice, ten groups each of five mice ($n = 5$), together with the glycolipid adjuvant C34[35,37] or with only phosphate-buffered saline (PBS), as a control, in a prime-boost regimen. Post-immune sera (day 42) were analyzed using glycan microarrays constructed by immobilizing dPNAG glycans **41–49** onto maleimide-coated glass slides; antibody binding was detected using Alexa Fluor 647 conjugated to goat anti-mouse IgG antibodies (Fig. 8A). Mice immunized with dPNAG-CRM197 conjugates of different dPNAG lengths and degrees of *N*-acetylation together with the glycolipid C34 adjuvant produced various anti-dPNAG IgG antibodies in different titers (Fig. 8B–E). Among the vaccine candidates that are non-acetylated

(compounds **55–58**), the dPNAG octamer-CRM197 conjugate (8mer-NH2-CRM197) (compound **56**) induced the highest IgG titer, and these antibodies recognized all the dPNAG glycans on the array using serum in 1:1000 to 1:8000 dilutions. Moreover, at a 1:8,000 serum dilution, the binding signals were 2–3 fold higher than produced by antisera induced by other non-*N*-acetylated dPNAG-CRM197 conjugates. The second highest IgG titer among the non-acetylated conjugates was induced by the tetramer (4mer-NH2-CRM197), followed by the dodecamer (12mer-NH2-CRM197) and finally the octadecamer (18mer-NH2-CRM197) (Fig. 8B, D). Interestingly, although the structures of dPNAGs found in biofilms of pathogens typically consist of more than 130 monomeric units[10], the longer glycoconjugates synthesized in our study did not induce the highest IgG titers.

For all the dPNAG-CRM197 conjugates we examined, the immune responses induced by conjugates of partially *N*-acetylated dPNAG did not exceed that induced by the non-*N*-acetylated 8mer-NH2-CRM197. The following tendency was found in terms of induced IgG titers: 8mer-NH2-CRM197 **56** > the CRM197-conjugate of the dPNAG dodecamer with 45% *N*-acetylation (12mer-45%NHAc-CRM197) **52** > 8mer-40%NHAc-CRM197 **51** > 4mer-NH2- CRM197 **55** > 12mer-60%NHAc-CRM197 **54** > 12mer-NH2- CRM197 **57** > 18mer-NH2-CRM197 **58** > 18mer-45% NHAc-CRM197 **53** > 4mer-45%NHAc-CRM197 **50** (Fig. S8). Notably, the dPNAG 4mer antigen apparently lost the ability to induce IgG as a result of even partial *N*-acetylation (Fig. 8B, D). Moreover, the dPNAG 4mer-45%NHAc glycan **41** was recognized by all antisera samples but at the lowest IgG titers compared to other dPNAG oligomers. The dPNAG dodecamer-CRM197 ($n = 11$) **52**, with 45% *N*-acetylation, induced significantly more IgG than did the non-acetylated dPNAG dodecamer-CRM197 **57** (Fig. S87). On the other hand, with the longer dPNAG octadecamers ($n = 17$) **53** (45% *N*-acetylated) and **58** (non-acetylated), the partial *N*-acetylation did not always increase IgG titers, despite their structure being closer to that of dPNAGs in microbial biofilm matrices.

## Opsonophagocytic killing of different bacteria strains

The functional activity of antibodies raised against the dPNAG-CRM197 conjugates (**50–58**) was assessed using an in vitro opsonophagocytic killing (OPK) assay (Fig. 9)[38,39]. We chose three prevalent nosocomial infection-causing pathogens for our study: *Staphylococcus aureus* (strain Newman), *Acinetobacter baumannii* (strain ATCC 17978), and *Streptococcus pneumoniae* serotype 19 A (clinical strain, MDR). These selections include both Gram-positive bacteria (*S. aureus* and *S. pneumoniae*) and Gram-negative bacteria (*A. baumannii*). We exposed the three pathogens to a mixture of DMF-treated differentiated HL60 cells, newborn rabbit complement, and antisera obtained from mice immunized with dPNAG-CRM197 conjugates. These mixtures were diluted in ratios ranging from 1:10 to 1:640. The results of the opsonophagocytic killing (OPK) assay indicated that the antisera from dPNAG-

**Fig. 5 | Synthesis of dPNAG glycans with thiol-functionalized antigens.** Reagents and conditions: (a) $H_2$, Pd(OH)$_2$/C, cat. AcOH, MeOH/CH$_2$Cl$_2$, RT, o/n; (b) Linker a, Et$_3$N, RT, o/n **28**: ($n$ = 3) 62%, **29**: ($n$ = 7) 75%, **30**: ($n$ = 11) 53%, **31**: ($n$ = 17) 51% (two steps); (c) N$_2$H$_4$·H$_2$O, EtOH, reflux, 3 h, **32**: ($n$ = 3) 93%, **33**: ($n$ = 7) 91%, **34**: ($n$ = 11) 96%, **35**: ($n$ = 17) 91%; (d) Ac$_2$O, NaHCO$_3$, H$_2$O/MeOH, RT, o/n **36**: ($n$ = 3), **37**: ($n$ = 7) **38**: ($n$ = 11), **39**: ($n$ = 17), **40**: ($n$ = 11); and (e) TFA, Et$_3$SiH, RT, 1 h, **41**: ($n$ = 3), **42**: ($n$ = 7), **43**: ($n$ = 11), **44**: ($n$ = 17), **45**: ($n$ = 11), **46**: ($n$ = 3) 95%, **47**: ($n$ = 7) 94%, **48**: ($n$ = 11) 91%, **49**: ($n$ = 17) 86%.

CRM197 immunized mice showed substantial bactericidal activity against all three pathogens. In accordance with established protocols, a reduction of more than 40% in bacterial colony-forming units (CFU) was considered indicative of significant killing activity[38].

The OPK assay results for activity against *S. aureus* showed that both the antisera raised against the non-acetylated 8mer-NH$_2$-CRM197 (**56**) and the 18mer-NH$_2$-CRM197 (**58**) exhibited substantial bactericidal activity within dilutions ranging from 1:10 to 1:40. In contrast, the antisera raised against the non-acetylated dodecamer 12mer-NH$_2$-CRM197 (**57**) displayed slightly less activity, but still had a 40% killing activity at a dilution of 1:320 (Fig. 9A). It is noteworthy that antisera raised against these non-*N*-acetylated conjugates (the 8mer-NH$_2$, 12mer-NH$_2$, and 18mer-NH$_2$) all had significant killing activities, extending to a dilution of 1:320. The antisera raised against the *N*-acetylated 8, 12, and 18mer conjugates all induced killing activities of ca. 40% up to a dilution of 1:80 (Fig. 9D). Contrary to the results from the glycan arrays, in which the octadecamer (18mer) conjugates were

found to induce lower IgG titers than the octamers (8mer) and dodecamers (12mer) (Fig. S8), in the OPK assay against *S. aureus*, the 18mers with or without *N*-acetylation gave similar results (Fig. 9A, D).

The OPK assay results for activity against *S. pneumoniae* showed that the antisera raised against the 8mer conjugate had higher killing activities than for the antisera raised against the 4- and 12mer conjugates up to a dilution of 1:80 (Fig. 9B). Antisera raised against the partially *N*-acetylated conjugates 8mer-40%NHAc- CRM197 and 12mer-45%NHAc-CRM197 resulted in higher killing activities that remained effective up to a dilution of 1:160 (Fig. 9E). In general, antisera raised against *N*-acetylated dPNAG conjugates had slightly higher OPK activities against *S. pneumoniae* (serotype 19 A) than their non-*N*-acetylated equivalents (Fig. 9B, E).

The OPK assay results for activity against *A. baumannii* showed the antisera raised against the non-acetylated dPNAG conjugates had significant killing activities up to a dilution of 1:40 (Fig. 9C). However, antisera raised against partially *N*-acetylated 4- and 8mer conjugates

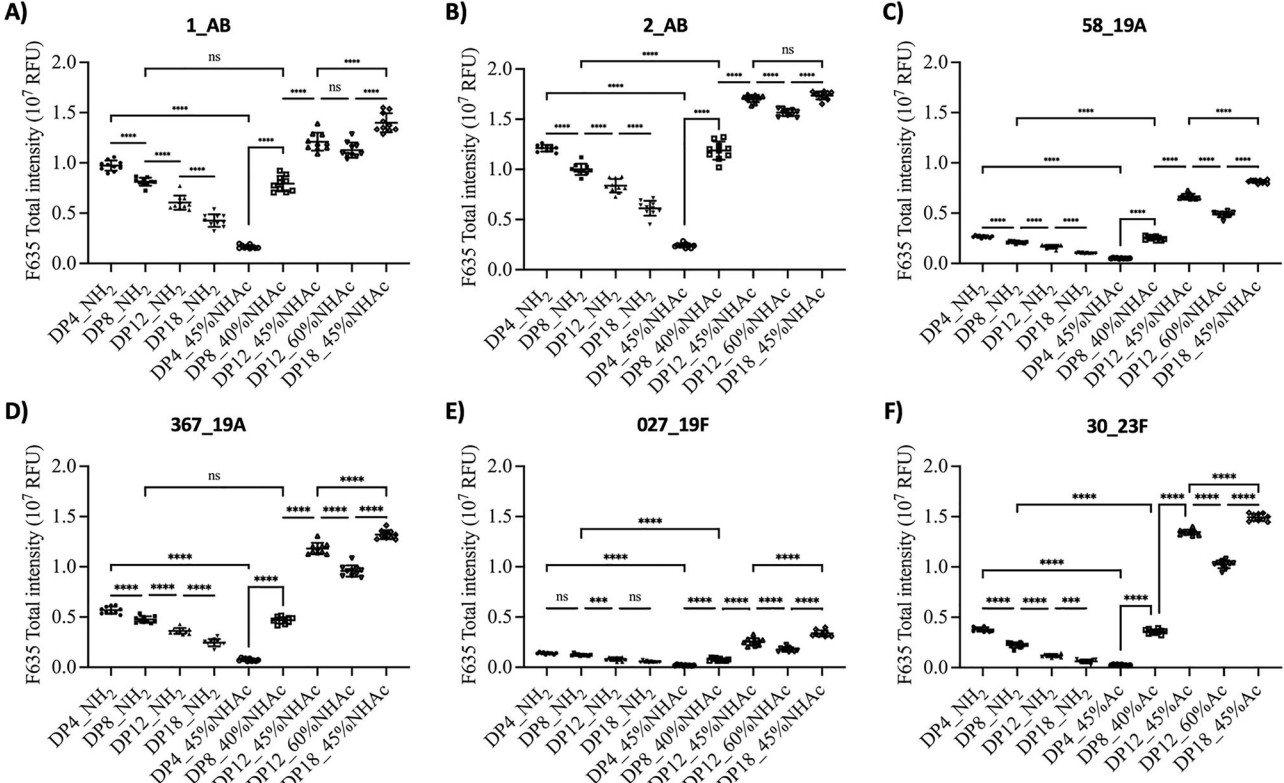

**Fig. 6 | Glycan array analysis of antibodies that bind to dPNAG glycans in sera from patients with bacterial infections. A**, **B** Sera from *Acinetobacter baumannii* infected patients. **C**, **D** Sera from a *Streptococcus pneumoniae* serotype 19 A infected patients. **E** Serum from a *Streptococcus pneumoniae* 19 F infected patient. **F** Serum from a *Streptococcus pneumoniae* 23 F infected patient serum. The negative control (BK) with the y-axis adjusted to accommodate its values is shown in Fig. S86 in

Supporting Information. The experiment was performed in technical replicates, and error bars represent the standard deviation from the mean of the data point ($n = 10$). Statistical analysis was performed with one-way ANOVA followed by Tukey's multiple comparison test using GraphPad Prism version 10.4.0 for macOS. Statistical significance and P values are shown in Table S2. ***: $p < 0.001$; ****: $p < 0.0001$; ns: not significant.

had considerably enhanced killing activities compared with equivalent non-acetylated 4- and 8mer conjugates, with significant killing activities up to a dilution of 1:160 (Fig. 9F). On the other hand, antisera raised against the *N*-acetylated 12mer gave only a slightly higher killing activity compared with antisera raised against the non-acetylated 12mer, with the OPK activities being higher than 40% up to a dilution of 1:80. Although antisera raised against the 4mer-45% NHAc-CRM197 gave very low IgG titers in the glycan arrays, their OPK activities against *A. baumannii* were high. Finally, antisera raised against the partially *N*-acetylated or non-acetylated 18mer were effective up to a dilution of 1:10, with somewhat lower killing activities shown by antisera raised against by the partially *N*-acetylated conjugate. This suggests that longer glycan chains may not result in better protection against *A. baumannii* despite their similarity to naturally occurring partially *N*-acetylated PNAG in biofilm matrices (Fig. 9F).

### In-vivo protection against Staphylococcus aureus

An in vivo study was conducted to evaluate the efficacy of the vaccine candidates in mice model challenged with *S. aureus* (Newman). Notably, all mice immunized with 8mer-40%NHAc-CRM197 (**51**) and 8mer-NH₂-CRM197 (**56**) exhibited 100% survival (Fig. 10). Similarly, mice vaccinated with 12mer-45%NHAc-CRM197 (**52**) demonstrated a strong survival rate, with the partially acetylated (45% Ac) variants generally providing superior protection compared to their non-acetylated counterparts (Fig. S88). Specifically, 12mer-60%NHAc-CRM197 (**54**) group, 60% mice survived after 4 days of bacterial inoculation, which is less than 12mer (45% Ac) with 80% survival. These findings indicate that partial acetylation is critical for eliciting protective immunity;

however, higher degrees of acetylation do not necessarily enhance antibody-mediated protection against *S. aureus* in vivo.

### Discussion

We have successfully synthesized a series of dPNAG glycans using an $n + 2$ strategy to elongate the glycan chains up to the octadecamer (18mer). To simulate dPNAG structures in biofilm matrices, the synthesized glycans were partially acetylated to reflect the heterogeneous pattern of acetylation found in bacterial systems. The potency of non-*N*-acetylated and partially *N*-acetylated dPNAG glycans as vaccine antigens against different bacteria was evaluated by conjugating them to a carrier protein, followed by glycan microarray analysis, resulting in the successful induction of antibodies against dPNAG. OPK experiments further corroborated the efficiency of the conjugates as vaccines against three bacterial pathogens.

Analysis by the dPNAG glycan microarrays demonstrated the effectiveness of three dPNAG conjugates: octamers (8 mers) with and without *N*-acetylation and a 45% *N*-acetylated dodecamer (12 mers), respectively. Antisera raised against the same conjugates had high killing activities in OPK assays against the Gram-positive pathogens *S. aureus* (Newman) and *S. pneumoniae* serotype 19 A. Moreover, the *N*-acetylated octamer (8 mer) conjugate was effective against the Gram-negative pathogen *A. baumannii*. Based on these results, we propose the most effective length of the dPNAG glycans to be approximately 8 to 12 monosaccharide residues. Furthermore, the OPK assays suggest that the partial *N*-acetylation of dPNAG glycans may induce antisera with broad-based immunity among bacterial pathogens. Although the non-acetylated octamer (8mer) and octadecamer (18mer) conjugates gave antisera with higher killing activities against *S. aureus* than antisera

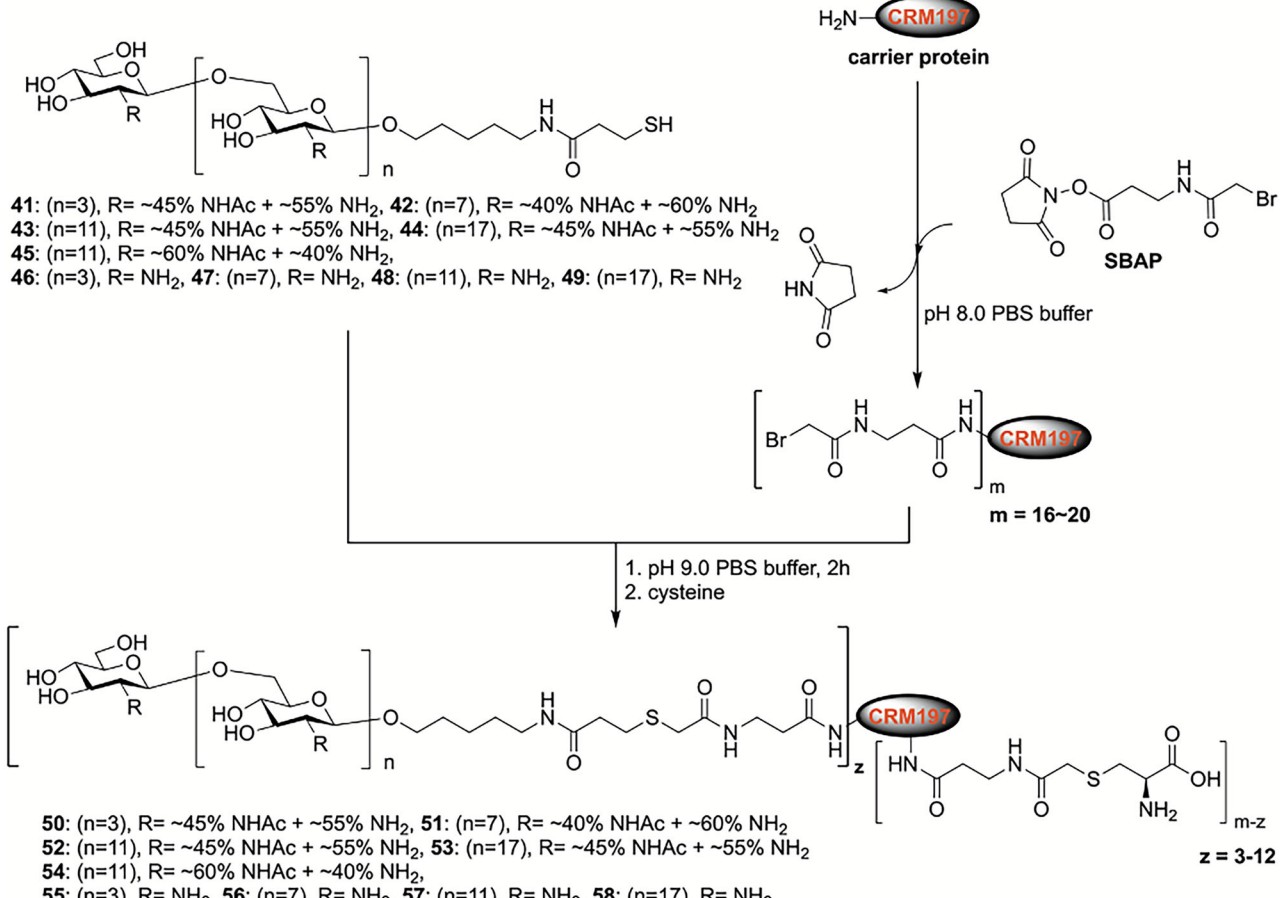

**Fig. 7 | Synthesis of dPNAG-CRM197 conjugates as vaccine candidates.** Synthetic dPNAG glycans 41–49 were conjugated to CRM197, a nontoxic mutant of diphtheria toxin widely used as a carrier protein in licensed human vaccines, to enhance immunogenicity and support T-cell-dependent antibody responses. The conjugation was carried out using succinimidyl 3-(bromoacetamido)propionate (SBAP) to introduce thiol-reactive 2-bromoacetyl groups onto CRM197, followed by covalent coupling to dPNAG glycans bearing sulfhydryl functionalities to form stable thioether linkages. Residual reactive groups were quenched with cysteine to yield the final dPNAG glycan-CRM197 vaccine candidates 50–58.

raised using the partially *N*-acetylated equivalent conjugates, the differences were negligible. Our study clearly indicates that dPNAG conjugates with a *N*-acetylation of under 50% are more desirable antigenic structures for vaccine development. This observation was further supported by a recent study demonstrating that selective acetylation on a PNAG pentamer vaccine candidate conferred stronger protection against MRSA challenge compared to its fully deacetylated counterpart[40].

The length of the dPNAG glycan chain also affects the IgG titers and killing activities of the antisera raised. Despite the longer glycans more closely resembling the structures of naturally occurring dPNAGs found in biofilm matrices, the octadecamers (18mers), which were the longest glycan conjugates in our study, failed to induce antisera with higher titers of IgG or higher OPK activities in all three bacterial models. Although the tetramers (4mers) were the easiest to synthesize and showed promising effectiveness in glycan microarray analyses, antisera raised against them had killing activities significantly lower than antisera raised against the octamer (8mer) and dodecamer (12mer) conjugates. In terms of chain length, the partially *N*-acetylated 8mer and 12mer conjugates appear to be the best candidates for vaccines.

Our findings align with and extend prior studies demonstrating that both the degree of polymerization (DP) and the *N*-acetylation of PNAG oligosaccharides are critical determinants of immunogenicity and protective efficacy. Early studies showed that partially or fully deacetylated PNAG fragments elicit more robust protective antibody responses than fully acetylated forms, indicating the immunodominance of deacetylated epitopes[11,14]. More recently, the formation of a comprehensive synthetic library of 32 structurally defined PNAG pentasaccharides has been reported, obtained by systematically varying the number and positions of *N*-acetyl groups[40]. Their data demonstrated that vaccine efficacy is highly dependent on specific acetylation patterns, with constructs such as PNAG10 and PNAG26, bearing internal *N*-acetylation at residues B and D, eliciting stronger protective responses than fully deacetylated or fully acetylated analogs[40]. Notably, in our study, the acetylated PNAG-8mer-40%NHAc and PNAG-12mer-45%NHAc constructs may recapitulate key features of such protective epitopes, as identified by the monoclonal antibody F598, which exhibited preferential binding to partially acetylated sequences[40]. Furthermore, our data suggest that the octasaccharide to dodecasaccharide length may offer an optimal compromise between epitope presentation and B cell receptor crosslinking, outperforming both shorter and longer analogs in inducing protective immunity. Longer chains may exhibit conformational heterogeneity or steric hindrance that limits effective antibody binding, whereas shorter chains may lack sufficient epitope density to stimulate durable immune responses[41].

In summary, our research has pinpointed the ideal length for dPNAG glycans in glycoconjugate vaccines, ranging from 8 and 12 monosaccharide residues, and with degrees of *N*-acetylation levels falling between 40% and 50%. In this study, we modified the dPNAG glycans with linkers that allowed them to be easily conjugated to

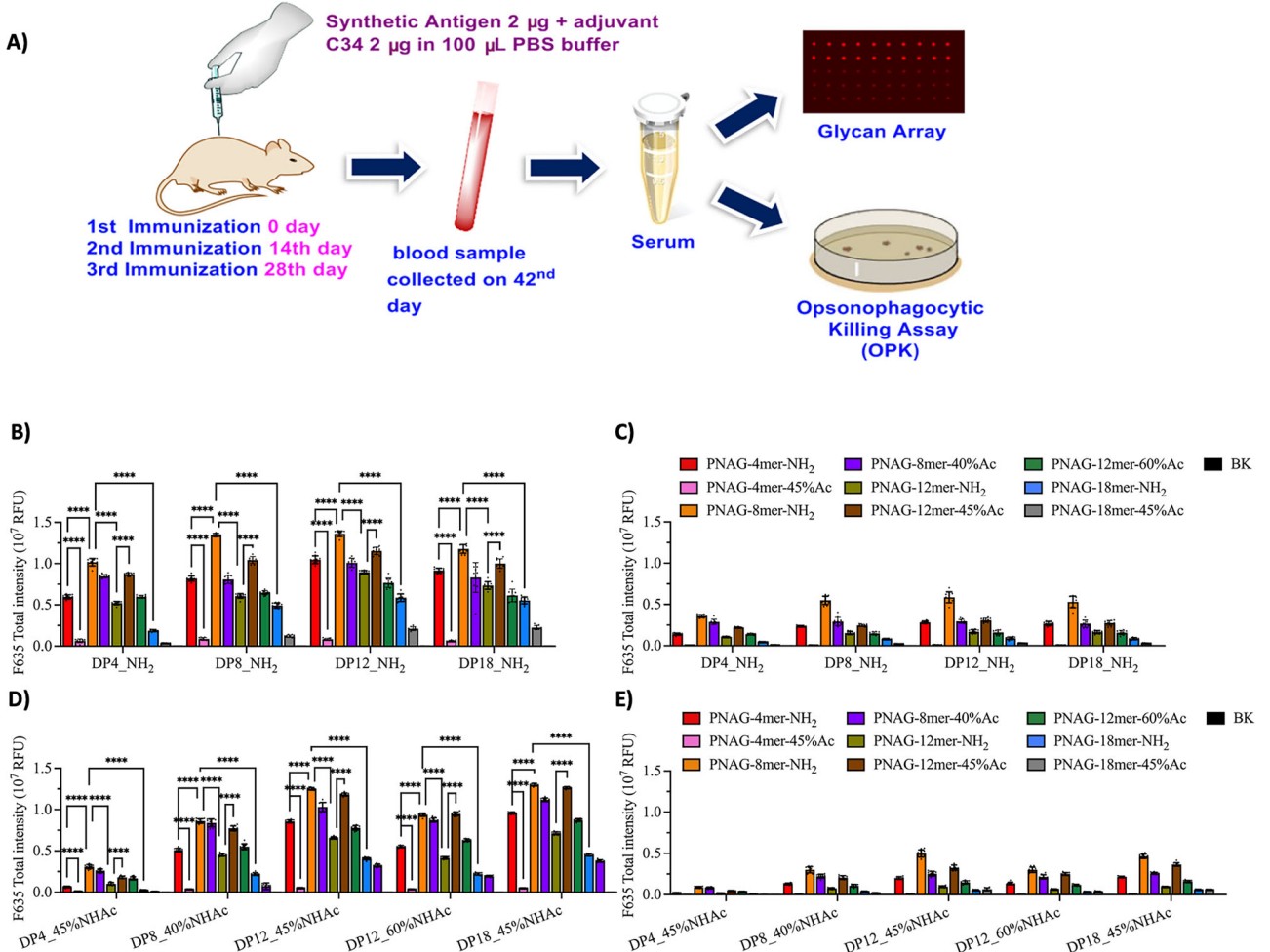

**Fig. 8 | Immunization of mice and evaluation of the immunogenicity of the dPNAG-CRM197 conjugates as vaccine candidates. A** Schematic diagram of the immunogenicity evaluation. **B**, **C** Glycan microarray with non-acetylated dPNAG glycans. **D**, **E** Glycan microarray with acetylated dPNAG glycans. Mice ($n = 5$) were immunized i.m. with the same amounts (2 mg) of the different dPNAG-CRM197 conjugates and glycolipid C34. Mouse sera were collected 14 days after the third immunization. The binding profiles of the antibodies with (**B**, **D**) 1000-fold or

(C and E) 8000-fold dilution were analyzed by glycan microarrays of the synthesized dPNAG glycans. The experiment was performed in biological replicates, and error bars represent the standard deviation from the mean of the data point ($n = 10$). Statistical analysis was performed with one-way ANOVA followed by Tukey's multiple comparison test using GraphPad Prism version 10.4.0 for macOS. Multiple comparisons and P values are shown in Table S3. ****: $p < 0.0001$.

carrier proteins and to be immobilized on microarrays, facilitating monoclonal antibody selection and binding analysis. Additionally, our $n + 2$ glycosylation approach enabled the synthesis of a diverse range of dPNAG glycans featuring desired degrees of *N*-acetylation, paving the way for the development of PNAG-based anti-bacterial vaccines.

## Methods

### Chemical synthesis of dPNAG-CRM197 conjugates

Full details of the synthesis of dPNAG-SH and product characterization are given in the Supplementary Information. To assemble dPNAG-CRM197 conjugates (**50**–**58**), CRM197-bromide was dissolved in PBS buffer (pH 9.0, 1.0 mg/mL, 2.0 mL), and then different dPNAG-SH oligosaccharides (compounds **41**–**48**) (2.0 mg/mL in PBS buffer, pH 9.0) were added into the solution. The mixtures were stirred at room temperature for 2 h then cysteine (1 mg) was added to quench the unreacted bromide functional groups. Unreacted dPNAG-SH and sodium phosphate salt were removed from the dPNAG-CRM197 conjugates by filtration [Amicon Ultra-0.5 (10 kDa)]. The dPNAG-CRM197 conjugates obtained were characterized using MALDI-TOF analysis to determine the carbohydrate incorporation rate. The final concentration of dPNAG-CRM197 is measured using a nanodrop, and the total

quantity of dPNAG glycan can then be calculated according to previously reported procedures[42,43]. The total glycan payload per carrier protein (i.e., the sugar-to-protein ratio) was estimated based on MALDI-TOF MS analysis. Specifically, the mass of dPNAG-SBAP-CRM197 was compared to that of SBAP-CRM197 alone, and the difference in their maximum mass peak values was attributed to the conjugated dPNAG glycans. Given the known molar concentration of the carrier protein and the calculated sugar-to-protein ratio, the total quantity of dPNAG glycans per CRM197 molecule could be determined. The unreacted dPNAG-SH can be recovered after reacting it with dithiothreitol (DTT), followed by purification using LH-20 column chromatography.

### Dosage and immunization schedule

To compare the immunogenicity of dPNAG derivative vaccines (dPNAG-CRM197 conjugates) (**50**–**58**) with PBS only, ten groups each of five *Mus musculus* (BALB/c strain; 6- to 8-week-old female, Bio-LASCO, Taiwan) were immunized intramuscularly with glycolipid C34 as an adjuvant. Three immunizations were given at 2-week intervals. Each vaccination contained dPNAG-CRM197 conjugate (2 μg) and glycolipid C34 (2 μg). Control mice were injected with phosphate buffer

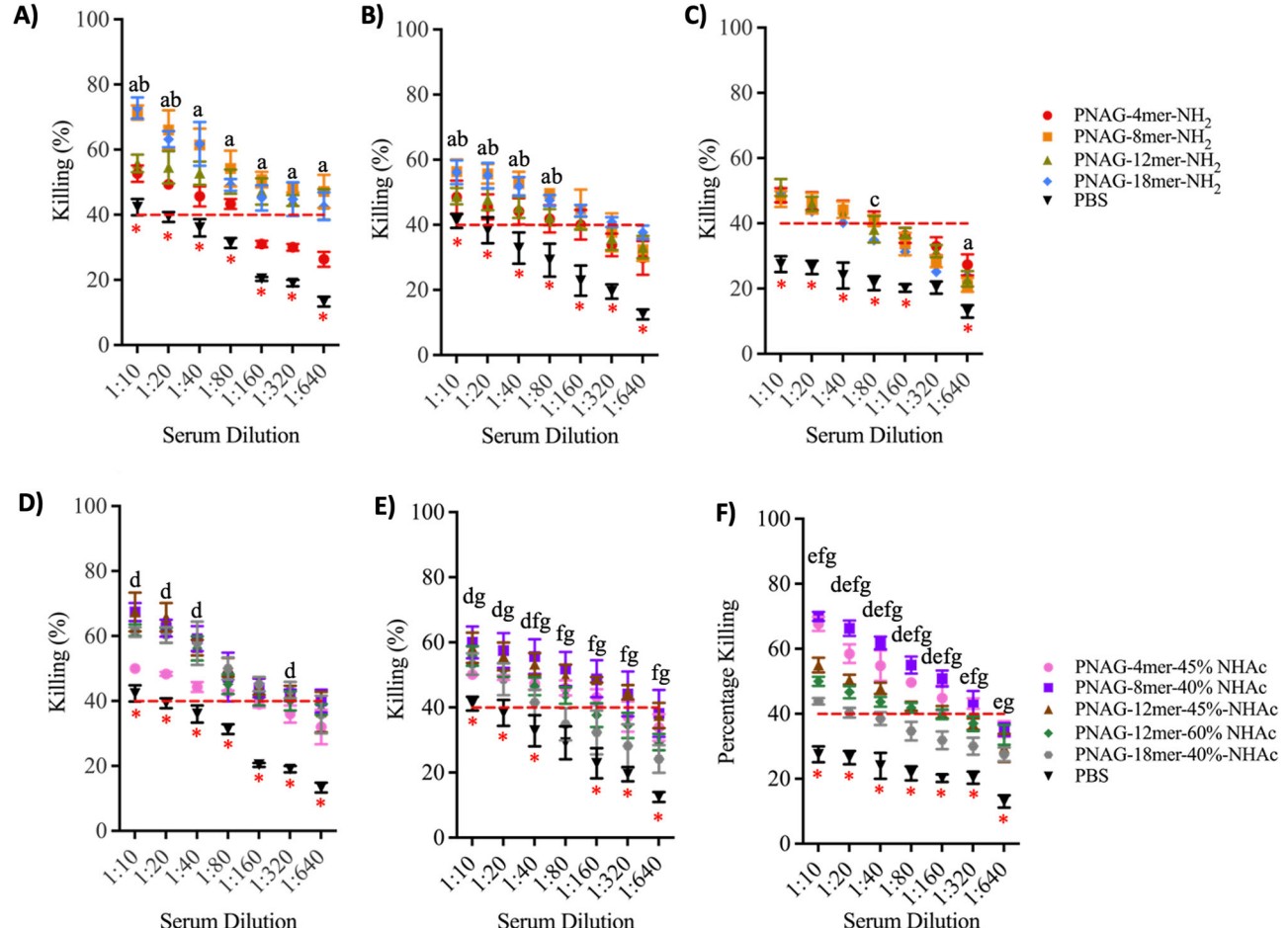

**Fig. 9 | Opsonophagocytic killing activities of mouse antisera on three species of bacteria following immunization with the dPNAG-CRM197 conjugates as vaccine candidates.** Mouse antisera obtained 14 days after the final immunization with dPNAG-CRM197 conjugates were diluted 1:10 to 1:640. The red dashed lines indicate an OPK activity of 40%. Comparison of the OPK activities of sera obtained from mice immunized with non-acetylated dPNAG 4, 8, 12, and 18mer-NH₂-CRM197 conjugates against (**A**) *S. aureus* serotype Newman, (**B**) *S. pneumonia* serotype 19 A, and (**C**) *A. baumannii* strain ATCC 17978; comparison of the OPK activities of sera obtained from mice immunized with partially *N*-acetylated dPNAG 4, 8, 12, and 18mer-NHAc-CRM197 conjugates against (**D**) *S aureus* serotype Newman, (**E**) *S. pneumonia* serotype 19 A, and (**F**) *A. baumannii* strain ATCC 17978. The experiment was performed in biological replicates, and error bars represent the standard deviation from the mean of the data point (with $n = 6$ for *S. pneumonia* serotype 19 A and *S. aureus* serotype Newman, and $n = 5$ for *S aureus* serotype Newman). Statistical analysis was performed with one-way ANOVA followed by Tukey's multiple comparison test using GraphPad Prism version 10.4.0 for macOS. Statistical comparisons of the killing at the same serum dilution, and with a $P < 0.05$, were defined as "a" for PNAG-8mer-NH₂ vs PNAG-4mer-NH₂, "b" for PNAG-8mer-NH₂ vs PNAG-12mer-NH₂, "c" for PNAG-8mer-NH₂ vs PNAG-18mer-NH₂, "d" for PNAG-8mer-40%Ac vs PNAG-4mer-45%Ac, "e" for PNAG-8mer-40%Ac vs PNAG-12mer-45%Ac, "f" for PNAG-8mer-40%Ac vs PNAG-12mer-60%Ac, and "g" for PNAG-8mer-40%Ac vs PNAG-18mer-40%Ac. The red asterisk indicates a significant difference between the negative control (PBS) with all other antisera tested at the same serum dilution in the assay. Multiple comparisons and P values are shown in Tables S4–S6.

saline (PBS). Mice were housed in individually ventilated cages under a 12 h light/dark cycle at 20–24 °C and 40–60% humidity, with ad libitum access to food and water. The use of a single sex was chosen to reduce variability related to hormonal cycles and to allow comparison with prior published data. Sex was therefore not a variable in the analysis. All procedures were approved by the Institutional Animal Care & Use Committee (IACUC) of Academia Sinica and conducted in accordance with ethical guidelines. The mice were bled before the first immunization (preimmune) and 10 days after the third immunization. The sera were obtained by centrifugation (4000 × *g*, 10 min), and. the serological responses were analyzed with the glycan microarrays.

### Construction of the dPNAG glycan arrays
To construct each glycan array, eight different dPNAG-SH glycans (compounds **41**–**48**) were first dissolved in the printing buffer (300 mM phosphate buffer, 0.005% Tween 20, pH 8.5) at a 10 mM concentration. These dPNAG-SH glycans were then printed (BioDot; Cartesian Technologies) using a robotic pin (SMP3; TeleChem

International) onto a maleimide-coated glass slide (ZeroBkg® surface; Microsurfaces Inc) with -0.6 nL of each of the different solutions deposited from a 96-well plate. Each microarray had 16 grids on one slide, with each grid consisting of 10 columns and 10 rows. The printed slides were allowed to react for 1 h in an 80% humidity chamber, dried overnight, and stored at room temperature in a desiccator prior to use.

### Serological studies of human sera using the dPNAG glycan arrays
The procedures for obtaining sera from patients infected with *A. baumannii, S. pneumoniae* serotype (serotype 19 F), *S. pneumoniae* (serotype 19 F), and *S. pneumoniae* (serotype 23 F) underwent a comprehensive review from Institutional Review Board (IRB) of Chang Gung Medical Foundation and were approved (IRB Number: 201902027A3D001). Before serum collection and the sharing of clinical data, all participants gave their informed consent. Out of the study participants, we selected six human serum samples, with two individuals infected with *A. baumannii* (patient code 1_AB and 2_AB; both

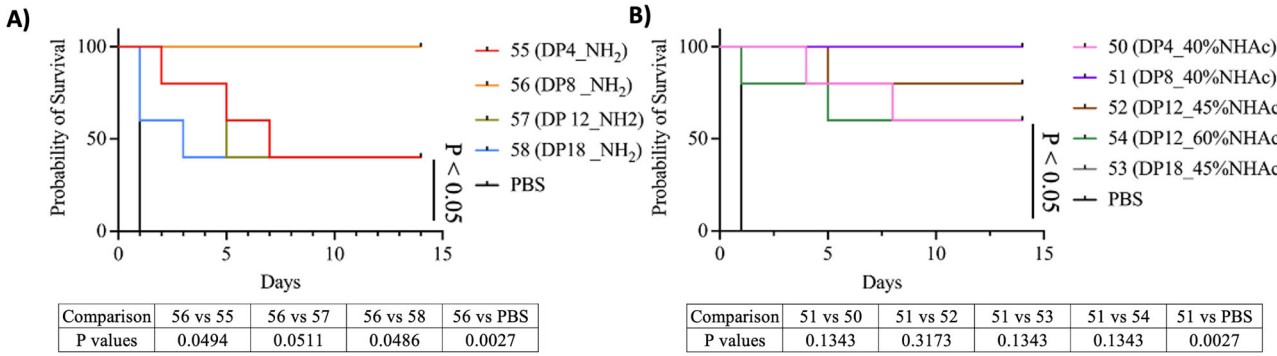

**Fig. 10 | Evaluation of in vivo efficacy of vaccine candidates for protection against *Staphylococcus aureus* infection. A** Evaluation of non-acetylated PNAG-4mer-NH₂ (DP4_NH₂, 55), PNAG-8mer-NH₂ (DP8_NH₂, 56). PNAG-12mer-NH₂ (DP12_NH2, 57), and PNAG-18mer-NH₂ (DP18_NH₂, 58); **B** evaluation of acetylated PNAG-4mer-45%NHAc (DP4_40%NHAc, 50), PNAG-8mer-40%NHAc (DP8_40% NHAc, 51), PNAG-12mer-45%NHAc (DP12_45%NHAc, 52), PNAG-12mer-60%NHAc (DP12_60%NHAc, 54), and PNAG-18mer-45%NHAc (DP18_45%NHAc, 53). Statistical analysis was performed using the two-sided Log-rank (Mantel-Cox) test in Kaplan-Meier survival analysis with GraphPad Prism version 10.4.0 for macOS. Comparisons were conducted pairwise; no multiple comparisons were performed.

male of undisclosed age), two infected with *S. pneumoniae* serotype 19 A (58_19A; female 4Y9M, and 367_19A; female 4Y6M), one infected with *S. pneumoniae* serotype 19 F (027_19F; male 5Y6M), and one infected with *S. pneumoniae* serotype 23 F (30_23F; female 6Y2M), based on their enzyme-linked immunosorbent assay (ELISA) antibody titers. Each serum sample was subjected to the following further treatments. First, the sera were diluted with 1% BSA/PBST buffer (PBST buffer: PBS and 0.05% Tween-20, pH 7.4) upon dilution factors. The glycan microarrays were blocked with Superblock blocking buffer (Pierce) for 1 h at 4 °C and washed three times with PBST buffer before use. The diluted sera were then added to the glycan microarrays and incubated for 1 h at 4 °C. Excess serum antibodies were washed away, and the microarrays were incubated in the dark for 1 h at 4 °C individually with the fluorescent dye Dylight™ 649 conjugated to donkey anti-human IgG (H + L chains) (Jackson ImmunoResearch Laboratories Inc., West Grove, Pennsylvania). The slides were then washed three times with PBST and scanned with a microarray fluorescence chip reader (GenePix 4300 A; Molecular Devices Corporation) using an excitation wavelength of 635 nm, and the scanned images were analyzed with GenePix Pro-6.0 analysis software (Axon Instruments, Union City, CA, USA).

### Serological studies of mouse antisera using the dPNAG glycan arrays
Sera from mice immunized with the different PNAG-CRM197 conjugates were treated in the same way as the human sera. Glycan microarray analysis was also done as described above for the human samples, except the secondary antibody was AlexaFluor 647-conjugated to goat anti-mouse IgG [H + L] (Jackson ImmunoResearch Laboratories Inc.) using an excitation wavelength of 635 nm, and the scanned images were analyzed with GenePix Pro-6.0 software. A bar graph displaying means with standard deviations was generated, and statistical comparisons between groups were performed using GraphPad Prism version 10.4.0 for macOS (GraphPad Software, Boston, Massachusetts, USA) to assess significance.

### Opsonophagocytic killing assay
Heat-inactivated sera samples from mice vaccinated with dPNAG-CRM197 conjugates and with PBS (as controls) were systematically diluted using two-fold serial dilutions in a 96-well microtiter plate containing Hanks balanced salt solution [HBSS] with calcium and magnesium, 0.1% gelatin, and 10% fetal bovine serum (FBS)[38]. The diluted suspensions were incubated with cells from various bacterial strains (approximately 104 CFU per well). A bacterial suspension (10 μL) was put into each well and incubated at room temperature for 60 min on an orbital shaker (50 rpm). Newborn rabbit complement (baby rabbit serum, final concentration 12.5%) and differentiated HL60 cells (ATCC CCL-240) were introduced at a 400:1 ratio to the bacteria (*S. aureus* serotype Newman, *S. pneumonia* serotype 19 A), and *A. baumannii* serotype 17978, respectively-complement-serum mixture (final volume 80 μL/well). The mixtures were then incubated in a tissue culture incubator (37 °C, 5% CO₂) for 90 min. Following incubation, the solution in each test well was diluted with an equal volume of 0.9% NaCl. A 10 μL aliquot was extracted and applied to a tilted blood agar plate, which was incubated at 37 °C with 5% CO₂ overnight. Control tubes with no serum and tubes containing normal rabbit serum (NRS) were included for comparison. The percentage killing was determined by calculating the ratio of the number of surviving CFUs in tubes with bacteria, differentiated HL60 cells, complement, and sera to the number of surviving CFUs in tubes lacking sera but containing bacteria, complement, and differentiated HL60 cells.

### Mouse challenge study
To assess efficacy against infection, a total of 45 healthy, 6-week-old female *Mus musculus* (BALB/c strain) were vaccinated with dPNAG glycan-CRM197 conjugates. Mice received three immunizations at 2-week intervals, with each dose containing 2 μg of dPNAG-CRM197 conjugate and 2 μg of glycolipid C34. Forty-two days after the initial immunization, all vaccinated mice (*n* = 45), together with control mice (*n* = 20) that received PBS injection were challenged with 2 × 108 CFU of *S. aureus* (Newman) and euthanized 14 days post-challenge. The use of a single sex was chosen to reduce variability related to hormonal cycles and to allow comparison with prior published data. Sex was therefore not a variable in the analysis. All procedures were approved by the Institutional Animal Care & Use Committee (IACUC) of Academia Sinica and conducted in accordance with ethical guidelines. Statistical analysis was performed using the two-sided Log-rank (Mantel-Cox) test in Kaplan-Meier survival analysis with GraphPad Prism version 10.4.0 for macOS. Comparisons were conducted pairwise; no multiple comparisons were performed.

### Reporting summary
Further information on research design is available in the Nature Portfolio Reporting Summary linked to this article.

## Data availability
The data that support the findings of this study are available within the article and its Supplementary Information and from the corresponding author(s) upon request. Source data are provided with this paper.

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

## Acknowledgements

This work was supported by the KTH Royal Institute of Technology, and the National Science and Technology Council, Taiwan (NSTC 114-2320-B-038-042-MY3) (NSTC 113-2636-M-038-001) (NSTC 104-2628-M-001-006) and Taipei Medical University (TMU109-AE1-B21). We thank Mrs. Yi-Ping Huang for assistance with the NMR analysis of the synthesized compounds. We also thank Mr. Chein-Hung Chen and Mrs. Chia-Lin Wu for their assistance with the mass analysis of the compounds, and Dr. Ya-Lin Lin for critical comments.

## Author contributions

K.L., C.Y.W. and Y.H. conceived and designed the project. K.L., M.K., T.M., M.H. conducted the experiments. K.L., M.K., T.C., B.I., P.J.H., J.S., C.C., Y.H. analyzed the data. K.L., B.I., P.J.H. and Y.H. wrote the manuscript. All authors read and approved the final manuscript.

## Funding

## Competing interests

The authors declare no competing interests.
