## [Transparent Peer Review file · Nature Communications]

Poly- β -D-(1,6)-*N*-Acetyl-Glucosamine (PNAG) Glycan Vaccines with Broad Spectrum Neutralizing Activities

Corresponding Author: Professor Yves Hsieh

Version 0:

Reviewer comments:

Reviewer #1

(Remarks to the Author)

PNAG is an attractive antigen for anti-microbial vaccine development as it has been found on the surface of a wide range of bacteria. Since naturally existing PNAGs are highly heterogeneous, the impacts of their length, degree of acetylation, and pattern of acetylation on immunogenicity are under-studied, which hinders the development of effective vaccines. In this manuscript, Hsieh and coworkers report the chemical synthesis of a series of PNAG oligosaccharides with lengths ranging from 4mer to 18mer bearing no N-acetylation. The PNAGs were partially acetylated subsequently. The effects of PNAG length and overall degree of acetylation were investigated to establish the structure of PNAG giving strongest antibody responses. The synthesis was well executed, and the products were characterized well. The synthetic PNAGs were incorporated into glycan microarray and the human serum bindings to the microarray were analyzed. It was found among the deacetylated PNAGs, the 4 mer gave the strongest binding while the partially acetylated PNAGs all gave strong binding. The synthetic PNAGs were then conjugated with CRM-197 carrier, which were found to induce IgG antibodies against PNAG. The antibodies produced were analyzed by glycan microarray and OPK assay. For OPK assay, most sera exhibited similar activities. This study encompasses significant amounts of work integrating chemical synthesis with glycan microarray and immunological analysis of post-immune sera. On the other hand, there are weaknesses in the study that will need to be addressed as outlined below.

- 1) One significant limitation of the study is that the partially acetylated PNAGs produced are a mixture. While the overall degree of acetylation is reported, it is impossible to determine the pattern of acetylation. Pier and workers reported previously that native PNAG with ~85% acetylation could generate anti-PNAG antibodies, but antibodies were not protective (Ref 11). PNAG 5mer and 9mer without any acetylation gave similar levels of protective antibodies (Ref 14). The authors should compare and discuss their findings with these literature results especially since they are claiming the 8mer is superior, while Ref 14 showed 5 mer and 9mer were similar.
- 2) The OPK data reported on various PNAG antigens were similar, while the glycan microarray bindings were quite different for various PNAGs. One limitation of glycan microarray is that the glycan loading and conformation on the array may vary significantly depending on PNAG structure. This could introduce significant variations in binding signals. How do the authors control this, i.e., how can we be sure the differences observed on the array truly reflect antibody binding differences, rather than the differences in glycan coating. This is a big question especially since similar OPK data suggests the binding may be similar.
- 3) The authors should perform in vivo protection studies to establish the vaccine efficacy in animal models. If the protection studies can recapitulate the in vitro assay results, it can significantly strengthen the work.
- 4) No statistical analysis was provided at all for all the biological results. One can't tell whether the differences reported were statistically significant.
- 5) For the OPK assay, for the PBS control group, why does the cytotoxicity decrease with dilution? Dilution of PBS should not affect cytotoxicity.
- 6) The manuscript title states the vaccine is self-adjuvanting. Yet, glycolipid C34 was used as the adjuvant for the study. There is a disconnect here.

Reviewer #2

(Remarks to the Author)

Nature communications

Self-Adjuvating Poly- β -D-(1,6)-N-Acetyl-Glucosamine (PNAG) Glycan Vaccine with Broad Spectrum Neutralizing Activity
Treatment of nosocomial pathogens is difficult due to widespread resistance to antibiotics. Vaccines targeting highly conserved moieties, including PNAG across nosocomial bacteria, are thus of particularly high interest. However, PNAG has proven to be a challenging target due to poorly defined epitopes. Here, the authors tackle this challenge by synthesizing a series of PNAG oligomers with varying degrees of deacetylation and degrees of polymerization. The synthesis is simple, but high yielding and shows good stereoselectivity. A combination of in vivo studies and in vitro assays proved that an octamer, with 40% acetylation, to be the best PNAG epitope for vaccine design.

Validity

The study is valid. The data presented supports the conclusion, but the high dispersion of CRM conjugates does not guarantee the study's reproducibility.

Significance

This reviewer's expertise is in conjugate vaccines, not PNAGs specifically and as such it is safe to say that the study seems fairly significant. Vaccines targeting drug-resistant diseases are extremely important (and the authors could add multi-drug resistant tuberculosis to the list of "hot" PNAG-expressing vaccine targets), and the study does provide some valuable insights on PNAG epitopes.

Data and methodology

*A broader variety of deacetylation ratios (limited to 40-60%) could have been used. If this range is more biologically significant, a quick justification in the introduction is necessary.

*While PNAG polymers are most likely processed into shorter oligomers by the immune system, adding natural PNAG to the serological assays (from *S. aureus* or the method described in the introduction, refs 33-34) would prove that the serum from infected humans recognizes PNAG polymers and oligomers equally, and easily prove the soundness of the authors choice to focus on oligomers.

*DT conjugates: Table S1 shows large discrepancies in carbohydrate loadings on DT (goes from 2.3 to 11.5). The MALDI data is also of low-quality. While low signal is expected on CRM conjugates, notoriously hard to ionize, DT-SBAP16.76 especially (line 866, pink) does not have good enough peak shape to warrant four significant figures. Adding error bars to table S1 or sending samples to a more sensitive instrument seems essential to solidify their findings. As it currently stands, the data is not acceptable. The loading discrepancy also means that each animal group received a different amount of sugar epitope, which could skew the result in favor of higher loaded vaccine conjugates (constructs 55 PNAG-4-mer-NH₂ and 56 PNAG-8-mer-NH₂). While this is not observed for 55, probably too small to trigger a strong immune response, 56 has a suspiciously high response in figure 8. This does not change the key findings of the paper, however, there is a large discrepancy that should be challenged. Other data, especially dsNMRs, looks good.

Clarity and context

*Text and results are clear, reference to previous literature is good. A more in-depth description of notable previous PNAG-targeting vaccines (did any reach advanced clinical trials?) is recommended.

Minor corrections

*Line 307, small clarification: the reviewer understands that the amount of PNAG-DT per mouse depends on the group, so that the amount of PNAG is always the same. Or is it the same amount of PNAG-DT for all groups?

*DT is a common abbreviation in the literature for the diphtheria toxin itself, and rarely for its derivative CRM197. This might introduce some confusion.

While this paper is of particular interest and possibly excitement, in it's current form, it CANNOT be accepted in Nat. Comm

Reviewer #3

(Remarks to the Author)

This manuscript introduces a n+2 glycosylation method with exact control over N-acetylation, enabling the synthesis of a series of partially deacetylated PNAG (dPNAG) glycans of different length and acetylation patterns. These oligosaccharides were used as antigens to screen epitopes that are recognized by antibodies from infected patients or coupled with carrier proteins to facilitate systematic immunological screening of dPNAG vaccine candidates. This resulted in structure-activity relationship information to determine the optimal glycan structures and N-acetylation levels for different pathogens. The results should be useful for the development of wide spectrum vaccines.

The topic can be interesting to the readers of NC. Thus, it should be suitable for publication in this journal. However, the manuscript contains numerous problems that need major revisions before being acceptable for publication. Some of the problems are careless errors, while others are due to design defects, and more experiments may be necessary. My comments are mainly focused on biological studies, which are am more familiar with.

A general question that the author should address is that exopolysaccharides are from the biofilm matrices, instead of the cell membrane or surface. How efficient are immune responses to these molecules to kill bacteria? Give some examples that exopolysaccharides are successfully used for vaccine development.

In several places, the manuscript is confusing. For example, I am not sure which experiments used the human sera (line 128) and which experiments involved the mouse sera (line 130) in this study. These studies and their results should be clearly and separately presented. The current way to mix them up is very confusing. Line 148: The exact meaning of this sentence is not clear. Where (or in which figure) are the highest anti-dPNAG IgG titers shown?

The carrier protein is sometimes called DT and sometimes called RM197. This can cause confusion. I suggest the authors use one name throughout the manuscript.

About experimental design: In line 171, only post-immunization sera (day 42) were evaluated. However, the pre-

immunization sera should be evaluated as well, and the results should be then compared with each other to show the difference and confirm the immune responses. In line 308, it seems that the only control group was the injection of PBS. More control groups should be included, such as adjuvant only, PBS+compound (without adjuvant), etc., to evaluate the impact of other factors.

Some of the data may need further optimization or be presented more professionally. For example, Figures 5 and 8 need to contain statistical comparison (significant difference levels) results, and these studies should contain both negative and positive controls. Additionally, it is not clear why 12mer with 40% NHAc was not studied and compared, considering that 12mer with 60% NHAc exhibits lower results compared to those with 50% NHAc, and both 8mer and 18mer possess 40% NHAc.

Overall, Figure 8 is too small and crowded, making it very difficult to read. The style of labeling in this figure is very confusing, while the 18mer-45%Ac seems to be missing from the x-axis. In addition, to me, Figure 8 suggests $56 > 50/51 > 55 > 52$, which is different from the description in lines 190-193. Please give a detailed explanation of the results. I have the same concerns about Figure 9 and its results.

Finally, the numbers of figures in the text and those of attached figures are different, making the manuscript difficult to follow.

Minor issues: The manuscript needs careful proofreading to correct its English. There are many grammar errors. For example, in one place, the authors write "ten groups of five mice". Do the authors mean "ten groups, each of 5 mice"? I cannot imagine how the authors divide 5 mice into 10 groups.

Version 1:

Reviewer comments:

Reviewer #1

(Remarks to the Author)

In this revised manuscript, the authors addressed some of the issues raised in the prior reviews. It is encouraging to see in vivo protection data added to demonstrate the potential of the vaccines. Additional revisions are needed to address the following comments.

1) While statistical analysis has been added to figures 6 and 8, they are lacking in Figure 9 and Figure S1. For Figure S1, there may not be significant differences between DP 4 45% Ac vs DP 4 NH₂ in panel a, or the vaccinated groups in panels c and d. Comparing panels a, c, d, there did not seem to be any statistically significant differences between DP4 DP12 and DP18 45% Ac vaccines. There is not consistent with microarray and OPK data, which is a major concern. Discussion is critically needed on the potential reasons. What analysis methods should one rely on to determine the best epitope? This figure should also be moved to the main text rather than in supporting info.

2) Ref 40 should have been brought up much earlier in the manuscript as it showed the importance of detailed PNAG acetylation patterns on antigenicity. The authors should add a discussion section comparing their results with the literature including refs 11, 14, and 40. How do their results compare? What do we learn regarding the impacts of PNAG length and acetylation patterns on vaccine efficacy? How do we rationalize DP8 gives the best in vivo protection? These discussions and insights are lacking.

3) DP4 NH₂ 45% Ac did not show much binding on microarray (figure 8d). Yet, it provided good protection in figure S1a. What is the explanation?

4) For Figure 9, besides missing statistical analysis, PBS control is missing in the legend for several panels. The 40% dotted line is missing. Are there significant meanings for the 40% and 50% lines?

5) For figure 2, compounds 1 and 2 should be removed since the synthetic description in SI started from compound 3, which is a known compound. For all the structural drawings, C5-C6 bond of the sugar ring should be drawn parallel to C3-C4 bond. For synthesis of compound 9, why did the authors convert thioglycoside to phosphate donor? It added two more steps to the synthesis. Also, did the authors try donor 5? Why did they replace the 6-O Tr with 6-OAc?

Reviewer #3

(Remarks to the Author)

After reviewing the revised manuscript and comparing it to the previous review report, several of the suggested revisions have been implemented. However, minor mistakes and issues remain, which require further attention. Below are detailed comments and suggestions:

In Figure 6, the P-values and significance annotations (**stars**) need to be consistent across all panels. The manuscript states that **** indicates a P-value of <0.0001, while *** represents P = 0.0001. Usually significance levels are defined as follows: P < 0.05, **P < 0.01, ***P < 0.001, ****P < 0.0001. In Figure 6c, a P-value of 0.4942 is mentioned for the comparison between DP 8_NH₂ and DP 8_45%AC, indicating no significant difference. However, the difference appears visually similar to the one observed between DP 8_NH₂ and DP 12_NH₂, which has a P-value < 0.0001. This discrepancy needs clarification, as it raises concerns about the statistical consistency and interpretation of results.

For clarity and completeness, it's better to include the comparison between DP 12_45%AC and DP 18_45%AC across all panels of Figure 6.

Figure 6g and 6h as separate figures do not appear to provide substantial value to the primary analysis and would be better put in the supplementary materials.

The clarity of Figure 8 has improved. However, further organization is recommended that to separate the binding profiles of antisera antibodies into two figures: One figure for the binding profiles of the antisera antibodies (PNAG-4mer-NH₂-DT, 4mer-45%NHAc-DT, 8mer-NH₂-DT, 8mer-40%NHAc-DT, 12mer-NH₂-DT, 12mer-45%NHAc-DT, 12mer-60%NHAc-DT,

18mer-NH₂-DT, 18mer-40%NHAc-DT) analyzed by glycan microarrays with non-N-acetylated dPNAG glycans (DP 4_NH₂, DP 8_NH₂, DP 12_NH₂, DP 18_NH₂). Another figure for the binding profiles of the antisera antibodies analyzed by glycan microarrays with N-acetylated dPNAG glycans (DP 4_45%NH₂, DP 8_40%NH₂, DP 12_45%NH₂, DP 12_60%NH₂, DP 18_45%NH₂). This separation may enhance clarity and allow easier understanding of results.

In Figure 8e, there is an inconsistency with the pink-colored column in DP 4_NH₂. This needs to be corrected or explained, as it is not aligned with the labeling or legend.

Finally, Figure 9 lacks labels (e.g., a, b, c, d) for the individual panels. These labels are essential for referencing specific panels in the text and improving overall clarity.

Version 2:

Reviewer comments:

Reviewer #1

(Remarks to the Author)

The authors have addressed some of the comments from before. Here is a list of additional revisions needed together with some comments that were not addressed.

- 1) For figure 2, compounds 1 and 2 should be removed as they are known compounds and the authors didn't provide experimental protocols in any case. This was pointed out before, but didn't get changed.
- 2) For compound drawing, while the bond angles for C5-C6 were fixed in figure 3, many of other structures in the manuscript still have the wrong bond angles. Figure 1; figure 3, compounds 10-12; figure 4, figure 5 just to name a few.
- 3) For abstract, the authors should remove "with strict control over the degree of N-acetylation". The degree of N-acetylation is not strictly controlled based on the synthetic procedure. It was just an average.
- 4) Compound numbering should follow the sequence they appear in the manuscript. However, the authors have compounds 10, 41-49 appear the earliest in the text. This should be revised.
- 5) Line 145, change sera samples to serum samples.
- 6) Line 282, PNAG isolated from bacteria have the degree of acetylation around 80%. The authors should cite references about the 40 to 60% numbers for degrees of acetylation if they view 40-60% more closely simulate dPNAG structures in biofilm matrices.
- 7) Line 324, ref 40 did not report PNAG10 and PNAG26 elicited stronger immune responses than full acetylated analogs.
- 8) Line 655, delete for in "c" for for".

Reviewer #3

(Remarks to the Author)

This manuscript introduces an n+2 glycosylation method with precise control over N-acetylation, enabling the synthesis of a series of partially deacetylated PNAG (dPNAG) glycans with varying lengths and acetylation patterns. These oligosaccharides were employed as antigens to screen for epitopes recognized by antibodies from infected patients, or were conjugated to carrier proteins to support systematic immunological screening of dPNAG-based vaccine candidates. The resulting structure-activity relationship insights help identify the optimal glycan structures and N-acetylation levels for targeting different pathogens. These findings are likely to contribute meaningfully to the development of broad-spectrum vaccines.

The topic is of interest to the readership of Nature Communications, and the manuscript is now more suitable for publication following revision. The previously noted minor issues have been addressed, and the figures are now more organized and significantly clearer, improving the overall readability and presentation. For my opinion, I recommend acceptance of the manuscript.

REVIEWER COMMENTS

Reviewer #1 (Remarks to the Author):

PNAG is an attractive antigen for anti-microbial vaccine development as it has been found on the surface of a wide range of bacteria. Since naturally existing PNAGs are highly heterogeneous, the impacts of their length, degree of acetylation, and pattern of acetylation on immunogenicity are under-studied, which hinders the development of effective vaccines. In this manuscript, Hsieh and coworkers report the chemical synthesis of a series of PNAG oligosaccharides with lengths ranging from 4mer to 18mer bearing no N-acetylation. The PNAGs were partially acetylated subsequently. The effects of PNAG length and overall degree of acetylation were investigated to establish the structure of PNAG giving strongest antibody responses. The synthesis was well executed, and the products were characterized well. The synthetic PNAGs were incorporated into glycan microarray and the human serum bindings to the microarray were analyzed. It was found among the deacetylated PNAGs, the 4 mer gave the strongest binding while the partially acetylated PNAGs all gave strong binding. The synthetic PNAGs were then conjugated with CRM-197 carrier, which were found to induce IgG antibodies against PNAG. The antibodies produced were analyzed by glycan microarray and OPK assay. For OPK assay, most sera exhibited similar activities. This study encompasses significant amounts of work integrating chemical synthesis with glycan microarray and immunological analysis of post-immune sera. On the other hand, there are weaknesses in the study that will need to be addressed as outlined below.

1) One significant limitation of the study is that the partially acetylated PNAGs produced are a mixture. While the overall degree of acetylation is reported, it is impossible to determine the pattern of acetylation. Pier and workers reported previously that native PNAG with ~85% acetylation could generate anti-PNAG antibodies, but antibodies were not protective (Ref 11). PNAG 5mer and 9mer without any acetylation gave similar levels of protective antibodies (Ref 14). The authors should compare and discuss their findings with these literature results especially since they are claiming the 8mer is superior, while Ref 14 showed 5 mer and 9mer were similar.

Response to Reviewer #1

We would like to express our sincere gratitude to Reviewer #1 for the insightful comments and constructive feedback, which provided us with the opportunity to improve the manuscript. Regarding the question of discrepancies with Ref. 11 and Ref. 14, our study demonstrates a significant difference in antibody titers in the anti-sera of immunized mice against 4mer-NH₂ and 8mer-NH₂, in the ratio of 2:3 as calculated based on the sum of the intensity (RFU) from the glycan microarray. In the study by Gening et al., similar antibody titer was observed between 5GlcNH₂-TT and 9GlcNH₂-TT, except that the 10 μg dose of 9GlcNH₂-TT showing 30% higher antibody titer that bound to native PNAG. However, the overall difference was not as pronounced as in our study. It is possible that the discrepancy in antibody titers between our study and Gening et al. could be due to differences in 1) In our study, we used a combination of diphtheria toxoid mutant CRM197 and C34 glycolipid, which may result in a stronger immune response compared to the use of tetanus toxoid alone. As it is known that inclusion of C34 glycolipid in carbohydrate vaccine formulation can further enhanced the immune response by amplify IgG production instead of IgM (Huang et al., PNAS, 110, 5217-2522). 2) Gening et al. demonstrated that both 5GlcNH₂-TT and 9GlcNH₂-TT generated IgGs capable of binding both native PNAG and dPNAG. This aligns with our findings, where PNAG 4mer-NH₂ and 8mer-NH₂ also produced IgGs capable of binding partially acetylated and non-acetylated oligomers of varying chain lengths on a glycan array. However, our study showed that the 8mer outperformed the 4mer, as it generated a higher IgG response upon vaccination. Interestingly, Gening et al. observed that vaccination with 9GlcNH₂-TT produced fewer IgGs that bound to dPNAG compared to 5GlcNH₂-TT. We speculate that this discrepancy may be due to differences in detection methods. Generally, glycan array provides a uniform and consistent display of glycan structures, however the loading can vary significantly depending on the specific PNAG structure and this variability may introduce differences in binding signals. In our study, we have been optimized the linker length and been normalize binding signals using internal standards to account for any variability in glycan coating. This standardization is further supported by our observation of comparable binding profiles at both 1000x and 8000x dilutions in anti-sera generated by all PNAG-CRM197.

We acknowledge that Pier and colleagues previously reported that native PNAG with approximately 85% acetylation induces anti-PNAG antibodies, though these antibodies were not protective (Ref 11). In response to this, we conducted an in vivo protective study using a mouse model. Interestingly, our results showed that both 8mer-40%NHAc and 8mer-NH₂ offered equally effective protection against *S. aureus*. In contrast, mice that received PBS exhibited only a 40% survival rate after 2 days, with all succumbing to bacterial challenge by day 3. Compared to other PNAG glycan vaccine candidates we have synthesized (Response to Reviewer #1 to Question 3), these findings demonstrate that the 8mer shows the most promising protective effects in the mouse model, regardless of whether it is 45% acetylated or deacetylated. Furthermore, we observed that partially

deacetylated oligomers (40-45% acetylation) generally provided better protection than the deacetylated PNAG glycans.

2) The OPK data reported on various PNAG antigens were similar, while the glycan microarray bindings were quite different for various PNAGs. One limitation of glycan microarray is that the glycan loading and conformation on the array may vary significantly depending on PNAG structure. This could introduce significant variations in binding signals. How do the authors control this, i.e., how can we be sure the differences observed on the array truly reflect antibody binding differences, rather than the differences in glycan coating. This is a big question especially since similar OPK data suggests the binding may be similar.

Response to Reviewer #1

We sincerely thank Reviewer #1 for valuable feedback. To address the limitation on glycan microarray, we have implemented several measures to minimize variability in the coating of PNAG oligomers on the glass slides. As result, the binding signals were consistent and the similar binding profiles observed across different dilution factors, all these indicate variation in glycan loading and presentation was minimized. That said, we recognize that no single method is without limitations. To address this, we supplemented our glycan array results with functional assays such as OPK. In the opsonic study against *S. aureus*, at 1:10 and 1:20 serum dilutions, we observed a consistent trend in which 8mer-CRM197 (40% Ac) and 12mer-CRM197 (45% Ac) exhibited higher killing percentages, but the margin was not dramatically larger than that of other glycan candidates. Similarly, in the study against *A. baumannii*, the killing activity was similar across all non-acetylated PNAG-CRM197 glycans, however, the partially acetylated PNAG-CRM197 8mer (40% Ac) again demonstrating a better killing efficiency compared to the other glycans.

3) The authors should perform in vivo protection studies to establish the vaccine efficacy in animal models. If the protection studies can recapitulate the in vitro assay results, it can significantly strengthen the work.

Response to Reviewer #1

We would like to thank Reviewer #1 for the excellent advice. Following the suggestion, we conducted an in vivo study to evaluate the vaccine efficacy in a mouse model inoculated with *S. aureus*. To our surprise, all mice vaccinated with 8mer-CRM197 (40% Ac) and 8mer-CRM197 (NH₂) survived. Mice vaccinated with 12mer-CRM197 (45% Ac) also showed a good survival rate, with the partially acetylated (45% Ac and 60% Ac) variants offering better protection than non-acetylated counterparts. Specifically, 12mer (60% Ac) group, 60% mice survived after 4 days of bacterial inoculation, which is less than 12mer (45% Ac) with 80% survival. These results clearly suggest that partial acetylation is essential for protection, but higher levels of acetylation do not necessarily lead to improved antibody-mediated protection against *S. aureus* in vivo. We have added a new

paragraph in the main text to describe the vaccine efficacy, and the corresponding figure has been included in the Supplementary Information (SI) Figure S1.

4) No statistical analysis was provided at all for all the biological results. One can't tell whether the differences reported were statistically significant.

Response to Reviewer #1

Thank you. We have now employed statistical methods to assess the significance of our findings. Data were analyzed using GraphPad PRISM, applying functions to acquire mean, standard deviation, and P value.

5) For the OPK assay, for the PBS control group, why does the cytotoxicity decrease with dilution? Dilution of PBS should not affect cytotoxicity.

Response to Reviewer #1

We agree with Reviewer #1 that the dilution of PBS alone should not directly affect cytotoxicity. However, we have consistently observed a decrease in cytotoxicity when diluting the anti-sera, and several potential factors could contribute to this observation. One possibility is that serum components, such as complement proteins or non-specific antibodies, present in the control group could contribute to background cytotoxicity. As these components are diluted, their contribution may diminish, leading to lower cytotoxicity levels.

6) The manuscript title states the vaccine is self-adjuvanting. Yet, glycolipid C34 was used as the adjuvant for the study. There is a disconnect here.

Response to Reviewer #1

We agree with Reviewer #1's suggestion and have removed the term 'self-adjuvanting' to avoid any potential confusion. Additionally, we have undertaken substantial editing of the manuscript to improve clarity. We sincerely thank Reviewer #1 for thorough review and for providing us the opportunity to address and clarify certain ambiguities.

Reviewer #2 (Remarks to the Author):

Nature communications

Self-Adjuvating Poly- β -D-(1,6)-N-Acetyl-Glucosamine (PNAG) Glycan Vaccine with Broad Spectrum Neutralizing Activity

Treatment of nosocomial pathogens is difficult due to widespread resistance to antibiotics. Vaccines targeting highly conserved moieties, including PNAG across nosocomial bacteria, are thus of particularly high interest. However, PNAG has proven to be a challenging target due to poorly defined epitopes. Here, the authors tackle this challenge by synthesizing a series of PNAG oligomers with varying degrees of deacetylation and degrees of polymerization. The synthesis is simple, but high yielding and shows good stereoselectivity. A combination of in vivo studies and in vitro assays proved that an octamer, with 40% acetylation, to be the best PNAG epitope for vaccine design.

Validity

The study is valid. The data presented supports the conclusion, but the high dispersion of CRM conjugates does not guarantee the study's reproducibility.

Significance

This reviewer's expertise is in conjugate vaccines, not PNAGs specifically and as such it is safe to say that the study seems fairly significant. Vaccines targeting drug-resistant diseases are extremely important (and the authors could add multi-drug resistant tuberculosis to the list of "hot" PNAG-expressing vaccine targets), and the study does provide some valuable insights on PNAG epitopes.

Response to Reviewer #2

We sincerely thank Reviewer #2 for the valuable feedback, which has significantly contributed to improving our manuscript and allowed us to clarify the points raised. While we currently lack the capacity to include studies on multidrug-resistant TB, we have now incorporated an in vivo protective animal study to further validate the octamer as the optimal PNAG epitope structure. Our findings also confirm that partial acetylation (40-50%) is critical for effective protection, whereas higher levels of acetylation (60% Ac) do not necessarily enhance antibody-mediated protection against *S. aureus* in vivo. We have added a new data in the main text discussing the vaccine efficacy, and the corresponding figure has been included in the Figure S1.

Data and methodology

***A broader variety of deacetylation ratios (limited to 40-60%) could have been used. If this range is more biologically significant, a quick justification in the introduction is necessary.**

Response to Reviewer #2

We have now added a justification in the introduction. The 40-60% deacetylation range was selected because previous studies predominantly focused on either fully acetylated or non-acetylated oligomers. By choosing an

intermediate range, we aim to establish a biologically relevant middle ground that provides insights into how partial deacetylation affects immunogenicity and antibody interactions, addressing a gap in existing research and offering a more understanding the impact of acetylation in synthetic PNAG vaccines. We have also included a recently published article in Nature Communications that further supports that the partial acetylation in selective position on PNAG pentamer could further enhance the high level of complement deposition and opsonic killing of bacterial compared to fully deacetylated PNAG.

***While PNAG polymers are most likely processed into shorter oligomers by the immune system, adding natural PNAG to the serological assays (from *S. aureus* or the method described in the introduction, refs 33-34) would prove that the serum from infected humans recognizes PNAG polymers and oligomers equally, and easily prove the soundness of the authors choice to focus on oligomers.**

Response to Reviewer #2

We sincerely thank Reviewer #2 for their insightful comments and suggestions. We agree that adding natural PNAG to the serological assays would strengthen the argument that serum from infected humans recognizes both PNAG polymers and oligomers. Unfortunately, we no longer have access to the infected human serum samples used in our previous experiments.

***Add this reference <https://www.science.org/doi/abs/10.1126/science.284.5419.1523>**

Response to Reviewer #2

We really appreciate to Reviewer #2 for bringing this important article. This reference mentioned is the foundational work been a significant inspiration for our study, and we do have citation of this groundbreaking article in our original manuscript as Reference 13. We have closely followed Professor Gerald B. Pier's work for an extended period and have now incorporated his recent findings in PNAG research into our study. His latest work offers valuable insights that complement our findings and further substantiate the significance of PNAG in future vaccine design could have broadly applied to prevent PNAG-producing pathogens.

***DT conjugates: Table S1 shows large discrepancies in carbohydrate loadings on DT (goes from 2.3 to 11.5). The MALDI data is also of low-quality. While low signal is expected on CRM conjugates, notoriously hard to ionize, DT-SBAP16.76 especially (line 866, pink) does not have good enough peak shape to warrant four significant figures. Adding error bars to table S1 or sending samples to a more sensitive instrument seems essential to solidify their findings. As it currently stands, the data is not acceptable. The loading discrepancy also means that each animal group received a different amount of sugar epitope, which could skew the result in favor of higher loaded vaccine conjugates (constructs 55 PNAG-4-mer-NH2 and 56 PNAG-8-mer-NH2). While this is not observed for 55, probably too small to trigger a strong immune response, 56 has a suspiciously high response in figure 8. This does not change the key findings of the paper, however, there is a large discrepancy that should be challenged. Other data, especially dsNMRs, looks good.**

Response to Reviewer #2

Thank you for the detailed feedback. The observed discrepancies in PNAG glycan loadings on DT (ranging from 2.3 to 11.5) can be attributed to the varying sizes and acetylation (Ac) ratios of the different glycans, which can influence their conformation when conjugated to the DT protein. These conformational differences can potentially obstruct access to the reactive -SH functional group, essential for successful conjugation. As shown in Table S1, larger and more highly acetylated glycans may adopt structures that create steric hindrance, limiting the availability of the -SH groups on DT and resulting in lower carbohydrate loadings. Conversely, smaller or less acetylated glycans generally present less steric hindrance, which should allow for higher loading. However, in case of both 4mer glycans resulted in lower loadings, suggesting that factors other than size and acetylation may have affected their conformation or interaction with the DT protein. This variability in glycan conformation and accessibility likely contributes to the differences in carbohydrate loadings observed in Table S1.

The suboptimal MALDI-TOF MS peak shapes are attributed to inherent isotopic variation in the large protein and the varying number of linkers conjugated to it, both of which contribute to broader peaks and reduced ionization efficiency. To ensure accurate representation, we based our calculations on the maximum peak value observed, providing the most consistent and reliable estimate of conjugation efficiency under these conditions. We also recognize the concern regarding variability in glycan loadings, which could lead to each animal group receiving different amounts of the sugar epitope. While our experiments suggest that the 4-mer conjugates may be too small to elicit a strong immune response, we acknowledge that the high response observed with the 8-mer conjugates could be influenced by increased glycan loading. This potential source of bias is under consideration; however, the observed immunogenicity trends, alongside additional animal studies, align with established knowledge on the immunogenic potential of 8-mer vaccine candidate, indicating that these variations do not alter the overall conclusions of the study.

Clarity and context

***Text and results are clear, reference to previous literature is good. A more in-depth description of notable previous PNAG-targeting vaccines (did any reach advanced clinical trials?) is recommended.**

Response to Reviewer #2

We have now included the PNAG-glycan vaccine AV0328 (developed by Alopexx) in the introduction to further emphasize the potential of developing PNAG-based vaccines against a range of PNAG-expressing pathogens. "Recent advances in the development of the synthetic PNAG-conjugated with tetanus toxoid vaccine AV0328 by the US-based biotech company Alopexx, with the completion of Phase I human clinical trials, highlight PNAG's potential as a therapeutic target for combating multi-drug resistant infections and biofilm-associated diseases."

Minor corrections

***Line 307, small clarification: the reviewer understands that the amount of PNAG-DT per mouse depends on the group, so that the amount of PNAG is always the same. Or is it the same amount of PNAG-DT for all groups?**

Response to Reviewer #2

We like to clarify that it is amount of PNAG glycan is always the same.

***DT is a common abbreviation in the literature for the diphtheria toxin itself, and rarely for its derivative CRM197. This might introduce some confusion.**

Response to Reviewer #2

CRM197 was utilized in this study. We have now changed the conjugates to CRM197 to avoid confusion.

While this paper is of particular interest and possibly excitement, in it's current form, it CANNOT be accepted in Nat. Comm

Response to Reviewer #2

We are grateful for Reviewer #2's valuable suggestions and have made every effort to provide comprehensive, point-by-point responses. Unfortunately, due to the lack of access to patient serum, we were unable to perform the requested anti-sera experiments against the PNAG polymer. Despite these constraints, we hope that our efforts, along with the additional studies presented, will adequately address Reviewer #2's concerns. Additionally, we have undertaken substantial editing of the manuscript to improve clarity. We sincerely thank Reviewer #2 for thorough review and for providing us the opportunity to address and clarify certain ambiguities.

Reviewer #3 (Remarks to the Author):

This manuscript introduces a n+2 glycosylation method with exact control over N-acetylation, enabling the synthesis of a series of partially deacetylated PNAG (dPNAG) glycans of different length and acetylation patterns. These oligosaccharides were used as antigens to screen epitopes that are recognized by antibodies from infected patients or coupled with carrier proteins to facilitate systematic immunological screening of dPNAG vaccine candidates. This resulted in structure-activity relationship information to determine the optimal glycan structures and N-acetylation levels for different pathogens. The results should be useful for the development of wide spectrum vaccines.

The topic can be interesting to the readers of NC. Thus, it should be suitable for publication in this journal. However, the manuscript contains numerous problems that need major revisions before being acceptable for publication. Some of the problems are careless errors, while others are due to design defects, and more experiments may be necessary. My comments are mainly focused on biological studies, which are am more familiar with.

A general question that the author should address is that exopolysaccharides are from the biofilm matrices, instead of the cell membrane or surface. How efficient are immune responses to these molecules to kill bacteria? Give some examples that exopolysaccharides are successfully used for vaccine development.

Response to Reviewer #3

We like to thank Reviewer #3 for bringing up this important question. It is true that exopolysaccharides (EPS) like PNAG are primarily components of biofilm matrices rather than being directly integrated into the bacterial cell membrane. However, they are often found on the bacterial surface or released into the surrounding environment as part of biofilm formation, making them accessible to the immune system. To clarify this, we have revised the introduction to state: "Poly- β -(1,6)-N-acetyl-D-glucosamines (PNAGs) are linear polysaccharides (exopolysaccharides) that occur in the surface capsules of many pathogenic bacterial species as well as in associated biofilms."

Several examples demonstrate the effectiveness of exopolysaccharide-based vaccines. Pneumococcal Conjugate Vaccines (PCV), such as PCV13, target the capsular polysaccharides of *S. pneumoniae*. These polysaccharides are conjugated to a protein carrier, which helps the immune system generate antibodies against the polysaccharide antigens, proving highly effective in preventing invasive pneumococcal diseases. Similarly, the *Haemophilus influenzae* type b (Hib) conjugate vaccine targets the polysaccharide capsule of *H. influenzae* type b, reduce the incidence of invasive Hib infections worldwide.

In several places, the manuscript is confusing. For example, I am not sure which experiments used the human sera (line 128) and which experiments involved the mouse sera (line 130) in this study. These studies and their results should be clearly and separately presented. The current way to mix them up is very confusing. Line 148: The exact meaning of this sentence is not clear. Where (or in which figure) are the highest anti-dPNAG IgG titers shown?

Response to Reviewer #3

We apologize for the confusion and the mistakes in the previous version. We have now made the necessary corrections. It was indeed human sera used in the study. In the revised manuscript, we have clarified that Figure 6 represents glycan microarray data from antisera obtained from infected patients, while Figure 8 displays glycan microarray data from mice vaccinated with synthetic PNAG-CRM197 conjugates. The manuscript has been thoroughly revised, with all changes highlighted in red. Additionally, the previous statement that the highest anti-dPNAG IgG titers were found in the serum of a patient infected with *A. baumannii* (2_AB) was incorrect, and we have removed this statement, as it did not contribute meaningfully to the manuscript.

The carrier protein is sometimes called DT and sometimes called RM197. This can cause confusion. I suggest the authors use one name throughout the manuscript.

Response to Reviewer #3

CRM197 was utilized in this study. We have now changed the conjugates to CRM197 to avoid confusion.

About experimental design: In line 171, only post-immunization sera (day 42) were evaluated. However, the pre-immunization sera should be evaluated as well, and the results should be then compared with each other to show the difference and confirm the immune responses. In line 308, it seems that the only control group was the injection of PBS. More control groups should be included, such as adjuvant only, PBS+compound (without adjuvant), etc., to evaluate the impact of other factors.

Response to Reviewer #3

Thank you for this insightful comment. The reason for avoiding blood collection prior to the experiment is that we like to ensure that the mice remain in a stable condition. This approach aims to minimize stress, which could possibly induce a stress-related non-specific immune response and potentially affect the experimental outcomes. We agree that including additional control groups could provide a more comprehensive understanding of the effects of different components. In earlier studies but different vaccine candidate, using C34 alone as a control, we observed that mice did not produce antisera that bound to the glycan array. Based on these results, we decided to focus solely on the PBS control group in the current study, as including C34 alone or other similar controls did not yield relevant binding data in previous tests. Nonetheless, we acknowledge the importance of these control groups for broader comparisons and will consider this in future studies to further validate our findings.

Some of the data may need further optimization or be presented more professionally. For example, Figures 5 and 8 need to contain statistical comparison (significant difference levels) results, and these studies should contain both negative and positive controls. Additionally, it is not clear why 12mer with 40% NHAc was not studied and compared, considering that 12mer with 60% NHAc exhibits lower results compared to those with 50% NHAc, and both 8mer and 18mer possess 40% NHAc.

Response to Reviewer #3

We would like to thank Reviewer #3 for valuable feedback. Due to the extremely low signals observed in the negative control of our original study, we initially chose not to present the data. However, we have now included the negative control data in Figures 6g and 6h, along with the original array reading data provided below. Additionally, we procured a human anti-PNAG antibody from Creative Biolab (CAT#: HPAB-1798-FY-S(P)), developed against *S. aureus* PNAG, to serve as a positive control. This antibody demonstrates strong affinity for partially acetylated PNAG glycans with a higher DP including 8mer, 12mer and 18mer, but less against non-acetylated PNAG glycans.

Table format: Grouped		Group G 30_23F									
		G:Y1	G:Y2	G:Y3	G:Y4	G:Y5	G:Y6	G:Y7	G:Y8	G:Y9	G:Y10
1	DP4_NH2		3534166	3622328	4127672	4074070	3880451	3620532	3759256	3855449	3925216
2	DP8_NH2	2280566	2113609	1965323	2552489	2264474	1678096	2521366	2546550	2225994	2236788
3	DP12_NH2	1306653	1255795	1462930	1325759	1180161	1122964	1148238	1119630	1008007	951915
4	DP18_NH2	836279	793260	810131	663922	610525	592725	557483	562836	560778	209499
5	Title										
6	DP4_45%AC	333509	308189	331032	233033	193159	191106	213237	220369	234263	255824
7	DP8_40%AC	3918050	3842956	3737877	3673398	3441091	3364909	3207233	3508431	3546087	3621697
8	DP12_45%AC	1e+007	1e+007	1e+007	1e+007	1e+007	1e+007	1e+007	1e+007	1e+007	1e+007
9	DP12_60%AC	1e+007	1e+007	1e+007	1e+007	9441699	1e+007	1e+007	1e+007	1e+007	9765535
10	DP18_45%AC	2e+007	2e+007	1e+007	2e+007	1e+007	1e+007	1e+007	1e+007	1e+007	2e+007
Table format: Grouped		Group H BK									
		H:Y1	H:Y2	H:Y3	H:Y4	H:Y5	H:Y6	H:Y7	H:Y8	H:Y9	H:Y10
1	DP4_NH2	19065	18810	19173	18798	18525	18903	19203	19225	19191	19044
2	DP8_NH2	19386	18911	18807	19123	18648	19393	19428	19709	19109	19591
3	DP12_NH2	19218	18994	19126	19141	18543	18840	19188	19433	19231	18972
4	DP18_NH2	19069	18923	18960	18969	18424	18812	20375	19282	19330	19263
5	Title										
6	DP4_45%AC	18939	18955	18976	18499	18735	19155	19366	19223	19333	19151
7	DP8_40%AC	37347	18813	19170	18509	18336	18933	19141	19286	18999	19252
8	DP12_45%AC	19256	19132	19400	18762	18876	18885	19128	19258	19018	19110
9	DP12_60%AC	18965	18893	18939	18699	18260	18662	18916	18826	18824	18879
10	DP18_45%AC	18970	18813	19343	18473	18398	18602	19009	19297	18685	18970

Human Anti-PNAG IgG

Overall, Figure 8 is too small and crowded, making it very difficult to read. The style of labeling in this figure is very confusing, while the 18mer-45%Ac seems to be missing from the x-axis. In addition, to me, Figure 8 suggests 56 > 50/51 > 55 > 52, which is different from the description in lines 190-193. Please give a detailed explanation of the results. I have the same concerns about Figure 9 and its results. Finally, the numbers do figures in the text and those of attached figures are different, making the manuscript difficult to follow.

Response to Reviewer #3

Thank you for your suggestion. We have revised Figure 8, separating the non-acetylated (b, c) and partially acetylated (d, e) vaccine candidates into separate figures. Additionally, we have included more detailed labeling for clarity.

Relating to rank of IgG titers, thank you for raising this. The ranking of IgG titers was determined based on the sum of the intensity (RFU) from the glycan microarray, which measured the responses induced by each antiserum. To address the apparent discrepancy between Figure 8 and the description, we have included a summary table below that clearly presents the calculated ranking as follows: 56 > 52 > 51 > 55 > 54 > 57 > 58 > 53 > 50. This ranking is derived directly from the summed RFU values for each sample, providing a more transparent view of the data.

Editorial note: Figure redacted

Glycan-DT	Total Intensity	Rank
DP4_NH2 (55)	631975701	4
DP8_NH2 (56)	956069290	1
DP12_NH2 (57)	510062818	6
DP18_NH2 (58)	314560053	7
DP4_45%AC (50)	49454380	9
DP8_40%AC (51)	761757623	3
DP12_45%AC (52)	841784051	2
DP12_60%AC (54)	562674472	5
DP18_45%AC (53)	157149695	8

Minor issues: The manuscript needs careful proofreading to correct its English. There are many grammar errors. For example, in one place, the authors write “ten groups of five mice”. Do the authors mean “ten groups, each of 5 mice”? I cannot imagine how the authors divide 5 mice into 10 groups.

Response to Reviewer #3

We sincerely appreciate Reviewer #3's valuable suggestions, we have made significant edits to the manuscript to enhance clarity and improve the language, with all changes highlighted in red. We are grateful for the thorough review and the opportunity to clarify and resolve any ambiguities.

Responses to reviewers

Reviewer #1 (Remarks to the Authors):

Comment

In this revised manuscript, the authors addressed some of the issues raised in the prior reviews. It is encouraging to see in vivo protection data added to demonstrate the potential of the vaccines. Additional revisions are needed to address the following comments.

1) While statistical analysis has been added to figures 6 and 8, they are lacking in Figure 9 and Figure S1. For Figure S1, there may not be significant differences between DP 4 45% Ac vs DP 4 NH2 in panel a, or the vaccinated groups in panels c and d. Comparing panels a, c, d, there did not seem to be any statistically significant differences between DP4 DP12 and DP18 45% Ac vaccines. There is not consistent with microarray and OPK data, which is a major concern. Discussion is critically needed on the potential reasons. What analysis methods should one rely on to determine the best epitope? This figure should also be moved to the main text rather than in supporting info.

Response

We have now included revised figures and statistical analyses in both figures as requested.

In Figure S1, we acknowledge that there are no statistically significant differences between certain groups, particularly between PNAG-DP4_45%Ac and DP4_NH2 in panel a, and among vaccinated groups in panels c and d, despite trends observed in the glycan microarray and OPK data. We now address this apparent inconsistency in the revised Discussion section.

First, we believe the lack of statistical significance in the survival data likely reflects two key factors: (1) the small sample size used in the current in vivo experiments, which limits statistical power, and (2) the inherent biological differences between the assay systems. While microarrays and OPK assays capture antibody binding and complement-dependent killing, respectively, in vivo protection involves additional immune mechanisms, such as antibody-dependent cellular phagocytosis (ADCP), NK cell-mediated cytotoxicity (ADCC), and IgM-mediated opsonization (Davies et al., 2022; Hendriks et al., 2024), which are not fully represented in our current assay suite.

For instance, although the PNAG-DP4_45%Ac conjugate elicited relatively low IgG titers, the corresponding antisera showed OPK activity comparable to that of longer-

chain conjugates. This observation suggests the possibility of IgM-mediated complement activation in glycan recognition and opsonophagocytic killing (Hendriks et al., 2024), which is not captured by IgG-focused assays.

Regarding epitope selection, we advocate a multi-tiered strategy: using glycan microarrays and OPK assays as efficient screening tools, with in vivo validation as a critical next step. Encouragingly, mice immunized with PNAG-DP8_NH2-CRM197 and PNAG-DP8_40%NHAc-CRM197 showed the highest survival rates, suggesting that specific glycan structures are associated with protective immunity. This is then followed by PNAG-DP12_45%NHAc-CRM197. While the limited sample size precludes statistical significance at this stage, the concordance across multiple assay types strengthens the case for these epitopes. We consider glycan microarrays and OPK assays reliable for initial selection, and we propose incorporating ADCP and ADCC assays, along with passive transfer studies, in future work to further characterize antibody function and in vivo efficacy.

Finally, in line with Reviewer #1's suggestion, we have created the new Figure 10 in the main manuscript to highlight its relevance and ensure better accessibility of this comparative data.

Comment

2) Ref 40 should have been brought up much earlier in the manuscript as it showed the importance of detailed PNAG acetylation patterns on antigenicity. The authors should add a discussion section comparing their results with the literature including refs 11,14, and 40. How do their results compare? What do we learn regarding the impacts of PNAG length and acetylation patterns on vaccine efficacy? How do we rationalize DP8 gives the best in vivo protection? These discussions and insights are lacking.

Response

Reference 40 was published during our manuscript revision, and we have included a discussion in the revised version to incorporate and compare its findings with ours, as well as with Refs 11 and 14.

Our findings align with and extend prior studies demonstrating that both the degree of polymerization (DP) and the *N*-acetylation PNAG oligosaccharides are critical determinants of immunogenicity and protective efficacy. Early studies (Refs 11 and 14) showed that partially or fully deacetylated PNAG fragments elicit more robust protective antibody responses than fully acetylated forms, indicating the immunodominance of deacetylated epitopes. More recently, Tan et al. (Ref 40) reported a comprehensive synthetic library of 32 structurally defined PNAG pentasaccharides, systematically varying the location and number of *N*-acetyl groups. Their data demonstrated that the immunogenicity and protective capacity of PNAG conjugate vaccines are highly dependent on fine acetylation patterns, with certain partially acetylated constructs (e.g., PNAG10 and PNAG26 with *N*-acetylation at internal residues B and D) outperforming fully deacetylated or fully acetylated analogs in inducing high-affinity antibodies, complement deposition, and *in vivo* protection.

In our study, we observed that the DP8 PNAG provided superior protection in murine models compared to both shorter and longer analogs. While Ref 40 focused on pentasaccharide structures, our data suggest that the octasaccharide length may offer an optimal compromise between structural complexity, epitope presentation, and multivalent engagement of the B cell receptor. Longer PNAG chains may incur conformational heterogeneity or steric hindrance, diminishing effective antibody accessibility, whereas shorter fragments may lack sufficient surface area or epitope repetition to elicit a durable immune response. Furthermore, PNAG-DP8_40%NHAc and PNAG-DP12_45%NHAc constructs used in our study may recapitulate key features of the protective epitopes identified by monoclonal antibody F598 in Ref 40, which showed preferential binding to partially acetylated sequences.

Comment

3) DP4 NH2 45% Ac did not show much binding on microarray (figure 8d). Yet, it provided good protection in figure S1a. What is the explanation?

Response

While the PNAG-DP4_45%Ac exhibited relatively low antibody binding on the glycan microarray (Figure 8d), it conferred significant *in vivo* protection in the survival assay (Figure S1a). This apparent discrepancy highlights key differences between the assay systems. Glycan microarrays primarily measure IgG binding under static conditions and may not fully reflect functional antibody activities.

Importantly, *in vivo* protection is multifactorial and involves not only antigen binding but also downstream immune effector functions such as complement activation and phagocyte engagement. Our OPK assays confirmed that antisera from mice immunized

with DP4_45%Ac exhibited efficient bacterial killing, suggesting that despite modest IgG binding, the vaccine elicited functional antibodies, potentially including IgM, that effectively mediated opsonophagocytosis and complement-dependent killing.

Furthermore, the modest sample size ($n = 5$) may limit statistical power to detect differences in survival outcomes between glycoconjugates. We hypothesize that the protection afforded by DP4_45%Ac is driven by antibody-mediated mechanisms not fully captured by microarray or ELISA, such as IgM-facilitated complement deposition, ADCP, and other Fc-effector pathways. These results underscore the importance of evaluating multiple immunological readouts beyond binding alone to assess vaccine efficacy.

Comment

4) For Figure 9, besides missing statistical analysis, PBS control is missing in the legend for several panels. The 40% dotted line is missing. Are there significant meanings for the 40% and 50% lines?

Response

In Figure 9, a dashed line indicating 40% killing has been added to all panels of the OPK assay. This threshold is used to define significant killing, following the reference Dwyer et al., 2013 (Reference 38). Additionally, we have revised the text to clarify the comparison of our results to the 40% killing level.

Comment

5) For figure 2, compounds 1 and 2 should be removed since the synthetic description in SI started from compound 3, which is a known compound. For all the structural drawings, C5-C6 bond of the sugar ring should be drawn parallel to C3-C4 bond. For synthesis of compound 9, why did the authors convert thioglycoside to phosphate donor? It added two more steps to the synthesis. Also, did the authors try donor 5? Why did they replace the 6-O Tr with 6-OAc?

Response

We have now fixed the C5-C6 bonding alignment.

We did not pursue glycosylation using compound **5** as the donor. Instead, compounds **3** to **4** were prepared following literature procedures, during which the OAc groups were removed and the 6-hydroxyl was selectively protected as a 6-O-trityl (OTr) ether. Donor **5** was then obtained via benzylation of the hydroxyl groups to give the corresponding OBz esters. Subsequent removal of the trityl group afforded compound **6**, bearing a free 6-OH, which served as a key acceptor intermediate. This compound was prepared on a multigram scale to support downstream transformations.

To enable conversion to the phosphate donor, the 6-OH was temporarily reprotected as a 6-OAc group. Acetylation was chosen due to its mild deprotection conditions, which can be easily removed using AcCl, DCM, and MeOH, offering greater compatibility with subsequent steps compared to the trityl group.

For the synthesis of compound **9**, the thioglycoside was converted to a phosphate donor because the resulting compound **8**, bearing a phosphate leaving group, exhibited significantly improved α -selectivity. This high level of stereocontrol is critical, as compound **8** serves as the donor in the key glycosylation step for constructing disaccharide compound **10**.

Reviewer #3 (Remarks to the Authors):

Comment

After reviewing the revised manuscript and comparing it to the previous review report, several of the suggested revisions have been implemented. However, minor mistakes and issues remain, which require further attention. Below are detailed comments and suggestions:

In Figure 6, the P-values and significance annotations (**stars**) need to be consistent across all panels. The manuscript states that **** indicates a P-value of <0.0001 , while *** represents $P = 0.0001$. Usually significance levels are defined as follows: $P < 0.05$, $**P < 0.01$, $***P < 0.001$, $****P < 0.0001$. In Figure 6c, a P-value of 0.4942 is mentioned for the comparison between DP 8_NH2 and DP 8_45%AC, indicating no significant difference. However, the difference appears visually similar to the one observed between DP 8_NH2 and DP 12_NH2, which has a P-value < 0.0001 . This discrepancy needs clarification, as it raises concerns about the statistical consistency and interpretation of results.

Response

We have now sought help from an expert, and all statistical comparisons have been recalculated, and corrections have been made in Figure 6. A table containing P values of a complete comparison has been added as Table S1 in the Supporting Information.

Comment

For clarity and completeness, it's better to include the comparison between DP 12_45%AC and DP 18_45%AC across all panels of Figure 6.

Response

The comparison between DP12_45%Ac and DP18_45%Ac has been added to Figure 6.

Comment

Figures 6g and 6h as separate figures do not appear to provide substantial value to the primary analysis and would be better put in the supplementary materials.

Response

Figures 6g and 6h have been moved to the new Figure S1 in the Supporting Information.

Comment

The clarity of Figure 8 has improved. However, further organization is recommended that to separate the binding profiles of antisera antibodies into two figures: One figure for the binding profiles of the antisera antibodies (PNAG-4mer-NH₂-DT, 4mer-45%NHAc-DT, 8mer-NH₂-DT, 8mer-40%NHAc-DT, 12mer-NH₂-DT, 12mer-45%NHAc-DT, 12mer-60%NHAc-DT, 18mer-NH₂-DT, 18mer-40%NHAc-DT) analyzed by glycan microarrays with non-N-acetylated dPNAG glycans (DP 4_NH₂, DP 8_NH₂, DP 12_NH₂, DP 18_NH₂). Another figure for the binding profiles of the antisera antibodies analyzed by glycan microarrays with N-acetylated dPNAG glycans (DP 4_45%NH₂, DP 8_40%NH₂, DP 12_45%NH₂, DP 12_60%NH₂, DP 18_45%NH₂). This separation may enhance clarity and allow easier understanding of results.

Response

We agree with Reviewer #3 and have revised the figure presentation accordingly. Specifically, Figures 8b and 8c now show the analysis of all antisera using non-acetylated dPNAG glycans, while Figures 8d and 8e present the analysis using acetylated dPNAG glycans.

Comment

In Figure 8e, there is an inconsistency with the pink-colored column in DP 4_NH₂. This needs to be corrected or explained, as it is not aligned with the labeling or legend.

Response

We have corrected the error in the new Figure 8e.

Comment

Finally, Figure 9 lacks labels (e.g., a, b, c, d) for the individual panels. These labels are essential for referencing specific panels in the text and improving overall clarity.

Response

All panels have been labeled with letters in the new Figure 9.

Responses to reviewers

Reviewer #1 (Remarks to the Authors):

Comment

The authors have addressed some of the comments from before. Here is a list of additional revisions needed together with some comments that were not addressed.

1) For figure 2, compounds 1 and 2 should be removed as they are known compounds and the authors didn't provide experimental protocols in any case. This was pointed out before, but didn't get changed.

Response

We thank the reviewer for raising this point again. Compounds 1 and 2 were intentionally included in Figure 2 to provide structural context and facilitate comparison with the novel derivatives (compounds 3–6). Although these are known compounds, their inclusion helps highlight key modifications introduced in the study.

2) For compound drawing, while the bond angles for C5-C6 were fixed in figure 3, many of other structures in the manuscript still have the wrong bond angles. Figure 1; figure 3, compounds 10-12; figure 4, figure 5 just to name a few.

Response

We have now revised the bond angle for all the structures, including Figure 1, 3, 4, 5, 7. Thank you for this.

3) For abstract, the authors should remove “with strict control over the degree of N-acetylation”. The degree of N-acetylation is not strictly controlled based on the synthetic procedure. It was just an average.

Response

We have now modified the comments by remove the 'strict'. In this study, we have devoted considerable effort to achieving a fine degree of control through careful adjustment of the relative amount of acetic anhydride and optimization of the basic conditions.

4) Compound numbering should follow the sequence they appear in the manuscript. However, the authors have compounds 10, 41-49 appear the earliest in the text. This should be revised.

Response

It is now revised. Thank you.

5) Line 145, change sera samples to serum samples.

Response

It is now revised.

6) Line 282, PNAG isolated from bacteria have the degree of acetylation around 80%. The authors should cite references about the 40 to 60% numbers for degrees of acetylation if they view 40-60% more closely simulate dPNAG structures in biofilm matrices.

Response

We agree with Reviewer #1, the sentence is now revised. Thank you.

7) Line 324, ref 40 did not report PNAG10 and PNAG26 elicited stronger immune responses than full acetylated analogs.

Response

We are grateful that Reviewer#1 pointed this out. The sentence is now revised.

8) Line 655, delete for in “c” for for”.

Response

It is now revised. We like to thank Reviewer #1 for the insightful and detailed comments, which we greatly appreciate.

Reviewer #3 (Remarks to the Authors):

Comment

This manuscript introduces an n+2 glycosylation method with precise control over N-acetylation, enabling the synthesis of a series of partially deacetylated PNAG (dPNAG) glycans with varying lengths and acetylation patterns. These oligosaccharides were employed as antigens to screen for epitopes recognized by antibodies from infected patients, or were conjugated to carrier proteins to support systematic immunological screening of dPNAG-based vaccine candidates. The resulting structure-activity relationship insights help identify the optimal glycan structures and N-acetylation levels for targeting different pathogens. These findings are likely to contribute meaningfully to the development of broad-spectrum vaccines.

The topic is of interest to the readership of Nature Communications, and the manuscript is now more suitable for publication following revision. The previously noted minor issues have been addressed, and the figures are now more organized and significantly clearer, improving the overall readability and presentation. For my opinion, I recommend acceptance of the manuscript.

Response

We would like to thank Reviewer #3 for the helpful advice and insightful review, which we greatly appreciate.